# SC²-WM: A Self-Correcting World Model with Closed-Loop Feedback for Vision-and-Language Navigation in Continuous Environments

Xuan Yao [1 2 3]    Yuze Zhu [2]    Junyu Gao [1 2]    Zongmeng Wang [1]    Changsheng Xu [1 2 4]

## Abstract

Vision-and-Language Navigation in Continuous Environments (VLN-CE) requires agents to make fine-grained navigation decisions under partial observability. However, most existing methods rely on open-loop execution, lacking mechanisms to detect and correct internal state drift during inference. We propose SC²-WM, a self-correcting world model framework that introduces internal feedback for closed-loop decision making in VLN-CE. Our method derives feedback from world-model foresight to perform state-level plan refinement before action execution. To handle challenging scenarios, we further introduce conditional world-aware adaptation, which enables model-level correction by selectively updating the world model at test time when feedback indicates model capacity insufficiency. Experiments on standard VLN-CE benchmarks demonstrate improved navigation robustness and generalization. Our code is available at https://github.com/sunrise-ikun/SC2_WM.

## 1. Introduction

Vision-and-Language Navigation (VLN) (Anderson et al., 2018b; Qi et al., 2020; Chen et al., 2022c) is a representative embodied AI task that requires an agent to navigate autonomously in 3D environments following natural language instructions. Targeting real-world interactive settings, this task captures tight coupling between language understanding and spatial decision making, making it a central topic in multimodal and embodied intelligence research (Zhang

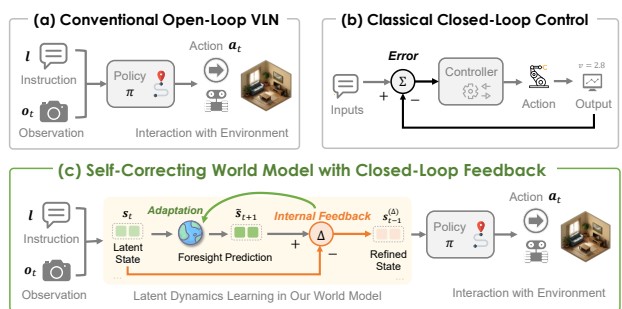

Figure 1. Comparison of (a) conventional open-loop VLN framework, (b) classical closed-loop control, and (c) our self-correcting world model with closed-loop feedback. Unlike open-loop methods that execute actions without validation, SC²-WM leverages internal feedback derived from foresight prediction to refine latent states before action execution.

et al., 2024; Gao et al., 2025b; Yuan et al., 2025).

Extending Vision-and-Language Navigation to continuous environments (VLN-CE) (Krantz et al., 2020; An et al., 2024) imposes substantially stricter demands on embodied agents. Beyond fine-grained control in a continuous action space, the agent must continuously update its understanding of the environment during execution. Moreover, real-world navigation is inherently partially observable, requiring the agent to integrate historical information into an internal representation while remaining aligned with the semantic constraints of natural language instructions.

Despite recent advances in model architectures and decision policies for VLN-CE (An et al., 2024; Wang et al., 2024b), most existing approaches still adopt an open-loop execution paradigm at inference time, as shown in Figure 1(a). The agent predicts actions from current observations and history, and executes them without validating their consequences. As a result, early decision errors may accumulate over time, degrading navigation performance. Classical closed-loop control (Figure 1(b)) addresses this by comparing outputs against references to compute error signals. While introducing closed-loop feedback mechanisms seems a natural solution (Bu et al., 2024; Li et al., 2024), doing so in VLN-CE is inherently challenging: external supervision is typically sparse and delayed, with success only

[1] State Key Laboratory of Multimodal Artificial Intelligence Systems (MAIS), Institute of Automation, Chinese Academy of Sciences (CASIA) [2] School of Artificial Intelligence, University of Chinese Academy of Sciences (UCAS) [3] HiThink Research, China [4] Peng Cheng Laboratory, ShenZhen, China. Correspondence to: Junyu Gao <junyu.gao@nlpr.ia.ac.cn>.

*Proceedings of the 43rd International Conference on Machine Learning*, Seoul, South Korea. PMLR 306, 2026. Copyright 2026 by the author(s).

observable at the trajectory level. Prior attempts incorporate feedback through reinforcement learning (Wang et al., 2024a) or model predictive control (Dey & Bhasin, 2025), but often rely on manually designed rewards and suffer from training instability. Test-time adaptation methods (Wang et al., 2020; Gao et al., 2024) instead adjust models using output-level statistics such as uncertainty or entropy, yet these signals remain decoupled from the agent's internal representation and cannot directly reveal misalignment in its understanding of the environment.

Based on the above analysis, we argue that effective closed-loop decision making in VLN-CE hinges on constructing appropriate feedback signals. Since external rewards are difficult to obtain or design reliably, we turn to an alternative: 'can the agent extract internal feedback signals from its own decision process for self-correction?' Predictive processing in cognitive science (Huang & Rao, 2011) suggests that adaptive behavior relies on continuously monitoring the consistency between internal predictions and subsequent observations, rather than on single-pass inference or delayed external supervision. This perspective aligns well with VLN-CE, where navigation failures often arise not from a single erroneous observation, but from gradual drift in the agent's internal understanding of the environment or instruction. If such drift can be detected during execution, the agent can correct its representation before errors amplify further.

Inspired by this, we propose SC$^2$-WM, a self-correcting world model framework for VLN-CE that enables closed-loop decision making through a computable internal feedback signal (Figure 1(c)). SC$^2$-WM focuses on detecting and correcting deviations in the agent's internal understanding of the environment during navigation. The core of SC$^2$-WM is a world model that captures environment dynamics and produces foresight predictions from latent state representations. These predictions reflect both the anticipated evolution of the environment and the agent's current internal assumptions. During inference, the discrepancy between the current latent state and its foresight provides an internal reference signal indicating how the selected action would steer the agent's internal dynamics before it is executed.

Based on this signal, SC$^2$-WM introduces a dual-level self-correction mechanism. *(i) At the state level*, feedback-guided plan refinement performs state-level correction by modulating the current latent state using foresight-derived signals, mitigating local inference drift during action selection. *(ii) At the model level*, conditional world-aware adaptation targets model-level correction by updating the world model when feedback reveals heavy reliance on foresight-derived guidance, improving its generalization ability for the testing environments. Together, these mechanisms enable effective navigation in continuous, dynamic environments.

Our main contributions are summarized as follows:

- We propose SC$^2$-WM, a dual-level self-correcting world model framework for VLN-CE that leverages internal foresight to enable closed-loop decision making with internal generated feedback.

- We introduce a dual-level self-correction mechanism, comprising feedback-guided plan refinement for immediate state-level correction and conditional world-aware adaptation for targeted model-level correction at test time.

- We conduct extensive experiments on standard VLN-CE benchmarks and real-world deployment, demonstrating that SC$^2$-WM consistently improves navigation performance.

## 2. Related Work

**Vision-and-Language Navigation in Continuous Environments.** Vision-and-Language Navigation (VLN) has emerged as a cornerstone of embodied AI, requiring agents to navigate unseen environments following natural language instructions (Anderson et al., 2018b; Qi et al., 2020; Banerjee et al., 2021; Chen et al., 2022b; An et al., 2022a; Qiao et al., 2023; Song et al., 2025; Guo et al., 2026a). While early works operated in discrete settings, Krantz et al. (2020) introduced Vision-and-Language Navigation in Continuous Environments (VLN-CE) to narrow the sim-to-real gap. This setting transitions agents from graph-based teleportation to continuous 3D spaces, introducing higher demands for understanding the environment state. To address the challenges posed by partial observability and the absence of topological priors in VLN-CE, many existing approaches adopt lightweight navigation architectures, improving navigation robustness and execution stability through enhanced state representation (Wang & Lee, 2025; Huang et al., 2025), trajectory modeling (An et al., 2024; Wang et al., 2023c), and action selection strategies (Wang et al., 2025c; 2024b). These methods typically emphasize efficient perception and compact policy learning, enabling practical deployment under limited computational resources while maintaining competitive navigation performance in complex environments. More recently, with the rapid advancement of multimodal large-language models (MLLMs), recent works (Lin et al., 2025; Zhang et al., 2025; 2024; Gao et al., 2025b; Yuan et al., 2025) have explored incorporating MLLMs into VLN to leverage their strong language understanding and cross-modal reasoning abilities, thereby enhancing the modeling of complex instructions, long-term dependencies, and high-level semantic relationships. Despite their promising semantic performance, such approaches typically incur substantial computational overhead and still face limitations in online inference efficiency and edge deployment. Therefore, we focus on the lightweight model paradigm for VLN-CE, investigating how to systematically improve navigation

decision-making and online adaptability within a compact model framework, without focusing on large-scale MLLMs.

**World Models.** World models are fundamentally defined as internal representations that capture environmental dynamics, enabling agents to simulate future states based on current observations and potential actions (Ha & Schmidhuber, 2018; Hafner et al., 2025). This predictive capability has been extensively applied across various embodied tasks, ranging from game simulation in virtual environments (Valevski et al., 2024) to autonomous driving (Ren et al., 2025; Russell et al., 2025) and robotic manipulation (Chi et al., 2025; 2024). In the context of VLN, world models empower agents to overcome partial observability through predictive foresight. DreamWalker (Wang et al., 2023b) integrates scene synthesis with Monte Carlo Tree Search (MCTS) for decision making. NavMorph (Yao et al., 2025) employs a recurrent state-space model to characterize the evolution of internal states. However, these approaches follow an open-loop paradigm, in which action execution is not guided by feedback signals, making them prone to the accumulation of navigation errors. Although a few works (Huang et al., 2025) explore predictive feedback, they are limited to modeling discrete environments and do not explicitly quantify error signals for joint correction at both the decision and model levels. To date, how to construct closed-loop, feedback-driven world models for tasks with sparse and delayed supervision remains an open problem.

**Closed-Loop Mechanisms for Embodied Tasks.** The goal of closed-loop control is to continuously adjust actions based on real-time sensory feedback to minimize the discrepancy between the current state and the target state (Hutchinson et al., 2002; Levine et al., 2016). Early approaches typically focused on tasks with real-time error sensors, such as visual servoing (Hutchinson et al., 2002). However, in contemporary embodied manipulation or navigation tasks, the absence of supervision signals often makes it difficult to compute error signals in real-time, hindering the construction of closed-loop systems. To address this, CLOVER (Bu et al., 2024) uses video diffusion models to generate visual sub-goals for establishing closed-loop control during manipulation. The fast-slow system paradigm (Li et al., 2024) leverages the semantic reasoning capability of large language models to construct feedback signals, where the fast system is responsible for efficient execution and the slow system is used for failure reflection and correction. Other methods have explored the design of feedback signals in VLA models, delegating closed-loop control to lightweight policies (Sendai et al., 2025). In recent years, world models have become a viable approach for building closed-loop feedback mechanisms due to their ability to predict future states. However, current explorations in this area are primarily based on the Model Predictive Control (MPC) framework (Sendai et al., 2025), which heavily relies

on reinforcement learning and is difficult to train. Therefore, this paper focuses on exploring how to efficiently extract internal feedback signals from the world model's own decision-making process to enable self-correction at both the decision and model levels.

## 3. Our Approach

In this section, we propose $SC^2$-WM, a self-correcting world model framework for VLN-CE. Our model incorporates an internal feedback mechanism to support closed-loop decision making, enabling self-correction to enhance navigation.

**Task Definition.** In VLN-CE (Krantz et al., 2020; Krantz & Lee, 2022; Wang et al., 2025c), an agent navigates in continuous 3D environments given RGB-D observations and a natural language instruction. Each episode starts from an initial position and terminates when the agent issues a 'STOP' action or reaches a predefined maximum number of steps. At each timestep $t$, the agent receives an observation $o_t$ and selects a navigation action $a_t$ based on predicted candidate waypoints (Krantz et al., 2021; Hong et al., 2022), which is then executed via low-level continuous control. Formally, the agent follows a learnable policy $\pi$ that maps the instruction $l$, observation history $o_{1:t}$, and past actions $a_{1:t-1}$ to the current action: $a_t \sim \pi(a_t \mid l, o_{1:t}, a_{1:t-1})$.

**Framework Overview.** As illustrated in Figure 2, we propose $SC^2$-WM, a self-correcting world model framework that integrates internal foresight with closed-loop feedback for robust decision making in VLN-CE. The framework supports two complementary forms of self-correction: *(i) feedback-guided plan refinement*, which performs immediate, state-level correction before action execution; and *(ii) conditional world-aware adaptation*, which selectively updates the world model at test time when feedback signals challenging scenarios, achieving model-level correction.

At each timestep $t$, the agent encodes the current observation $o_t$ into visual features $v_t$ and integrates them with the linguistic representation extracted from the instruction $l$ (Chen et al., 2022c; An et al., 2024). These features are then incorporated into a latent state $s_t$, maintained by the world model to capture the navigational context. This state is updated recurrently via a posterior transition $q(s_t \mid \cdot)$ that integrates the previous state, the executed action, and the current visual-linguistic features. Based on $s_t$, a foresight policy $\tilde{\pi}$ proposes a provisional action $\tilde{a}_t$, and the world model anticipates a foresight state $\tilde{s}_{t+1}$ via a prior transition $p(\tilde{s}_{t+1} \mid \cdot)$ without observing new visual input. An internal feedback signal derived from this foresight is used for plan refinement, correcting the current state to $s_t^{(\Delta)}$ (detailed in Section 3.1), from which the navigation policy $\pi$ selects the final action $a_t$. We formalize $SC^2$-WM as follows:

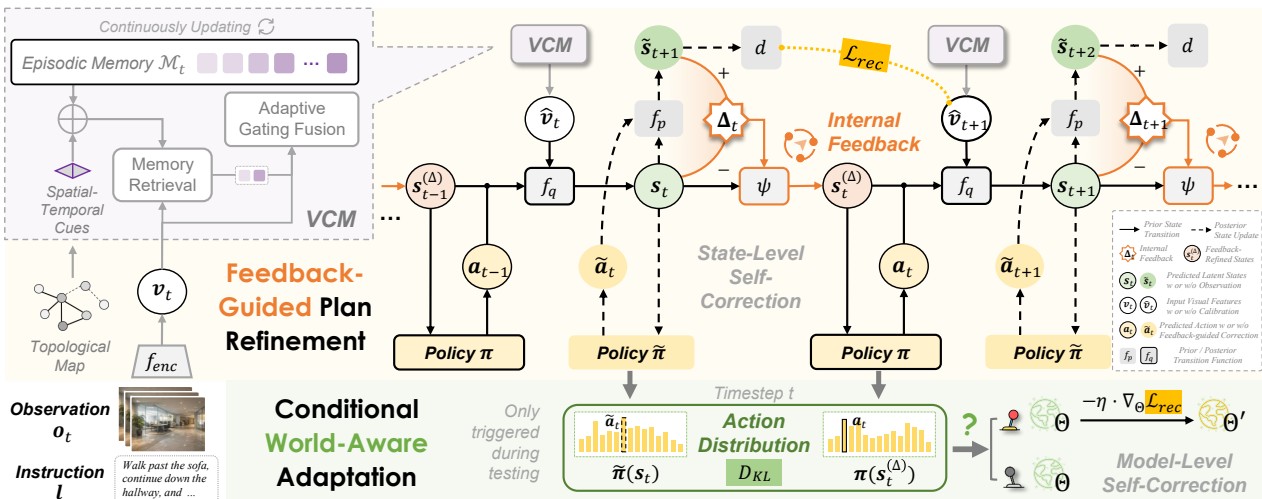

*Figure 2.* **Overview of SC²-WM framework,** which has two complementary self-correction modules: (i) feedback-guided plan refinement for state-level correction before action execution, and (ii) conditional world-aware adaptation for model-level correction during test time.

Visual Representation: $\boldsymbol{v}_t = f_{\text{enc}}(\boldsymbol{o}_t),$     (1)

Initial Latent State: $\boldsymbol{s}_0 \sim \mathcal{N}(\mathbf{0}, \mathbf{I}),$     (2)

Posterior State Update: $\boldsymbol{s}_t \sim q\big(\boldsymbol{s}_t \,|\, \boldsymbol{s}_{t-1}^{(\Delta)}, \boldsymbol{a}_{t-1}, \boldsymbol{v}_t, \boldsymbol{l}\big),$     (3)

Provisional Decision: $\tilde{\boldsymbol{a}}_t \sim \tilde{\pi}(\cdot \,|\, \boldsymbol{s}_t),$     (4)

Prior State Transition: $\tilde{\boldsymbol{s}}_{t+1} \sim p\big(\tilde{\boldsymbol{s}}_{t+1} \,|\, \boldsymbol{s}_t, \tilde{\boldsymbol{a}}_t\big),$     (5)

Action Prediction: $\boldsymbol{a}_t \sim \pi(\cdot \,|\, \boldsymbol{s}_t^{(\Delta)}),$     (6)

Visual Decoder: $\tilde{\boldsymbol{v}}_{t+1} \sim p\big(\tilde{\boldsymbol{v}}_{t+1} \,|\, \tilde{\boldsymbol{s}}_{t+1}\big),$     (7)

Here, the visual decoder reconstructs future visual features from the foresight state, providing a learning signal for training the world model, as described in Section 3.2.

When the above feedback-guided plan refinement leads to a pronounced change in the action distributions generated by $\tilde{\pi}(\boldsymbol{s}_t)$ and $\pi(\boldsymbol{s}_t^{(\Delta)})$, it suggests that foresight-derived guidance plays a substantial role in shaping the current decision. In such cases, SC²-WM selectively activates conditional world-aware adaptation to further enhance its modeling capacity for the current environment dynamics. This adaptation operates at the model level by updating the world model parameters using its internal self-supervised objective. We detail the adaptation procedure in Section 3.3.

### 3.1. Feedback-Guided Plan Refinement

Having introduced the overall pipeline, this section focuses on how the internal feedback signal is constructed and used for plan refinement. In our latent-based world model, an effective feedback mechanism critically depends on the quality of the latent state representation. To serve as a reliable basis for feedback, the latent state should capture action-

relevant dynamics while suppressing variations from viewpoint changes, partial observability, and rendering noise. Otherwise, the state shift induced by foresight may become ambiguous, making it unclear whether the predicted change reflects the agent's decision or incidental perceptual fluctuations. To mitigate this, we first calibrate the visual representation so that the latent state more faithfully reflects action-conditioned dynamics.

**Visual Calibration Module (VCM).** VCM maintains an episodic memory of recent visual representations, $\mathcal{M}_t = \{(\boldsymbol{k}_i, \boldsymbol{m}_i)\}_{i=t-L}^{t-1}$, where $L$ denotes the memory horizon. Each memory element $\boldsymbol{m}_i$ stores visual features extracted from past observations. To retrieve relevant context for calibration, VCM constructs attention keys on the fly by augmenting the stored visual features with relative spatial and temporal cues with respect to the current step $t$ (see Appendix for details).

Given the current visual representation $\boldsymbol{v}_t$ as the query, VCM retrieves contextual information from $\mathcal{M}_t$ via cross-attention with learnable temporal decay:

$$\boldsymbol{c}_t = \sum_i \text{softmax}(A_i)\,\boldsymbol{m}_i, \quad A_i \propto \boldsymbol{v}_t^\top \boldsymbol{k}_i - \varphi(\delta t_i), \quad (8)$$

where $\delta t_i = t - i$ is the temporal distance and $\varphi(\cdot)$ down-weights temporally distant observations.

The retrieved context $\boldsymbol{c}_t$ is then fused with the current representation via adaptive gating: $\hat{\boldsymbol{v}}_t = \boldsymbol{\alpha}_t \odot \boldsymbol{v}_t + (1 - \boldsymbol{\alpha}_t) \odot \boldsymbol{c}_t$, where $\boldsymbol{\alpha}_t$ is a learnable weight.

By aggregating observations from multiple viewpoints along the trajectory, the calibrated representation $\hat{\boldsymbol{v}}_t$ captures richer geometric and semantic cues about the environment, effectively mitigating ambiguity arising from partial ob-

servability and perceptual noise (Arandjelovic et al., 2016). This multi-view integration provides a more stable basis for latent-state inference, ensuring that state shifts induced by foresight primarily reflect action-relevant dynamics rather than incidental perceptual fluctuations.

**Internal Feedback.** With the calibrated visual input providing a stable latent state $s_t$, which better captures action-conditioned dynamics, we now construct an internal feedback signal from foresight to guide plan refinement. Based on $s_t$, the foresight policy $\tilde{\pi}$ proposes a provisional action $\tilde{a}_t$. Since future observations are unavailable before execution, we leverage the world model's prior transition $f_p$ to anticipate the outcome: $\tilde{s}_{t+1} = f_p(s_t, \tilde{a}_t)$. Here, $f_p$ is implemented as a Transformer-based transition model. Note that while the world model formulation in Eq.(5) models the prior transition as a distribution, we use the deterministic mean prediction (Min et al., 2024) to obtain $\tilde{s}_{t+1}$, avoiding the introduction of additional noise into the feedback signal.

The foresight state $\tilde{s}_{t+1}$ encodes the anticipated consequence of the provisional action. However, it represents a future latent state rather than actionable information for the current decision. We therefore extract the action-induced change by computing the latent discrepancy:

$$\Delta_t = \tilde{s}_{t+1} - s_t. \quad (9)$$

Notably, $\Delta_t$ is not a training target to be minimized, but a model-internal guidance signal that captures how the provisional action would steer the latent dynamics.

Based on this guidance, we perform self-correction by updating the current latent:

$$s_t^{(\Delta)} = \psi(s_t, \Delta_t), \quad (10)$$

where $\psi(\cdot)$ is a learnable neural network and $s_t^{(\Delta)}$ is the feedback-refined state for final action prediction (Eq.(6)).

The overall forward process of the proposed state-level self-correction mechanism is summarized in Algorithm 1.

### 3.2. Pre-training Objective

The pre-training of SC$^2$-WM aims to learn an internal latent state $s_t$ that is suitable for feedback-guided plan refinement. Specifically, the learned state is expected to (i) support reliable action prediction and (ii) encode predictive latent dynamics that anticipate future observations. Together, these properties provide the necessary foundation for feedback-guided refinement mechanism introduced in Section 3.1 and conditional world-aware adaptation described in Section 3.3.

We first impose action-level supervision on both original latent state $s_t$ and feedback-refined latent state $s_t^{(\Delta)}$, ensuring a consistent decision basis before and after refinement:

$$\mathcal{L}_{ac} = -\mathbb{E}_t\Big[\log p(a_t^* \mid s_t) + \log p(a_t^* \mid s_t^{(\Delta)})\Big], \quad (11)$$

---

**Algorithm 1** Feedback-Guided Plan Refinement *(Forward Pass)*

---

**Input:** Observation sequence $\mathcal{O}$, instruction $l$, navigation graph

**Output:** Executed action sequence $\mathcal{A}$, refined latent states

1: **Initialize:** episodic memory $\mathcal{M}_0 = \emptyset$, refined initial state $s_0^{(\Delta)}$
2: **for** $t = 1$ to $T$ **do**
3:     *// Visual Context Memory*
4:     $v_t = f_{enc}(o_t, l)$
5:     $m_t = \text{MemoryRetrieval}(\mathcal{M}_{t-1}, v_t)$
6:     $\hat{v}_t = \text{AdaptiveGatingFusion}(v_t, m_t)$
7:     *// Posterior state update*
8:     $s_t = f_q(s_{t-1}^{(\Delta)}, a_{t-1}, \hat{v}_t)$
9:     *// Foresight prediction*
10:    $\tilde{a}_t \sim \tilde{\pi}(\cdot \mid s_t)$
11:    $\tilde{s}_{t+1} = f_p(s_t, \tilde{a}_t)$
12:    *// Internal feedback and refinement*
13:    $\Delta_t = \tilde{s}_{t+1} - s_t$
14:    $s_t^{(\Delta)} = \psi(s_t, \Delta_t)$
15:    *// Action execution and memory update*
16:    $a_t \sim \pi(\cdot \mid s_t^{(\Delta)})$
17:    Execute $a_t$ and observe $o_{t+1}$
18:    Update $\mathcal{M}_t$ with $(\hat{v}_t, s_t^{(\Delta)})$
19: **end for**

---

where $a_t^*$ denotes the expert action at time step $t$. In practice, this objective is implemented using standard imitation learning supervision (An et al., 2024; Wang et al., 2025c).

In addition, we regularize the prior transition (Eq.(5)) by aligning it with the observation-conditioned posterior (Eq.(3)) via KL divergence, encouraging distributional consistency between inference and foresight:

$$\mathcal{L}_{wm} = \mathbb{E}_t\Big[D_{KL}\big(q(s_t \mid \cdot) \,\|\, p(\tilde{s}_t \mid \cdot)\big)\Big], \quad (12)$$

However, distributional alignment alone does not guarantee that the foresight state captures meaningful environment dynamics. To this end, we further supervise the action-conditioned prior transition by its ability to anticipate future visual features. Specifically, given the foresight state $\tilde{s}_{t+1}$ predicted by the world model, a lightweight decoder $d(\cdot)$ maps it to an estimate of the next-step front-view visual representation (Eq.(7)). The supervision signal $v_{t+1}$ is obtained by encoding the observation at $t+1$ with a pretrained visual encoder (Dosovitskiy, 2020):

$$\mathcal{L}_{rec} = \mathbb{E}_t\Big[\|d(\tilde{s}_{t+1}) - v_{t+1}\|_2^2\Big]. \quad (13)$$

Although defined in the visual feature space, this objective directly constrains the transitioned latent state, encouraging the world model to encode task-relevant environment dynamics without resorting to pixel-level reconstruction.

The final training objective jointly optimizes action prediction and predictive dynamics by combining navigation and world model losses: $\mathcal{L} = \mathcal{L}_{\mathrm{ac}} + \mathcal{L}_{\mathrm{wm}} + \mathcal{L}_{\mathrm{rec}}$.

### 3.3. Conditional World-Aware Adaptation

The feedback-guided refinement enables immediate self-correction by modulating the latent state before action execution, leveraging foresight from the trained world model. Its effectiveness thus depends on how accurately the world model captures the current environment dynamics.

During deployment, the agent may encounter environments whose visual appearance or dynamics differ from training. Under such domain shifts, the prior transition (Eq.(5)) learned offline may become less aligned with the current observations, reducing the informativeness of the resulting feedback signal. Adapting the world model to better reflect environment-specific characteristics can improve foresight quality, thereby strengthening feedback-guided refinement. We therefore introduce a conditional world-aware adaptation mechanism that selectively updates the world model parameters at test time, activated only when the agent's decision relies heavily on feedback guidance.

To determine when adaptation is beneficial, we monitor the magnitude of feedback-induced corrections via the KL divergence between action distributions before and after refinement: $\kappa_t = D_{\mathrm{KL}}(\tilde{\boldsymbol{p}}_t \,\|\, \boldsymbol{p}_t)$, where $\tilde{\boldsymbol{p}}_t = \mathrm{softmax}(\tilde{\pi}(\boldsymbol{s}_t))$ and $\boldsymbol{p}_t = \mathrm{softmax}(\pi(\boldsymbol{s}_t^{(\Delta)}))$. A large $\kappa_t$ indicates that feedback has substantially altered the agent's action preference, suggesting that the world model would benefit from adapting to the current environment dynamics.

Instead of relying on a fixed threshold, we maintain a running buffer of recent $\kappa$ values and trigger adaptation when $\kappa_t$ exceeds the $\tau$-th percentile of this empirical distribution. We further require that the entropy of $\boldsymbol{p}_t$ falls below a threshold $\epsilon$, ensuring that updates occur only when the corrective signal is both substantial and confident.

Once triggered, we update the world model using the self-supervised reconstruction objective $\mathcal{L}_{\mathrm{rec}}$ (Eq.(13)):

$$\boldsymbol{\Theta} \leftarrow \boldsymbol{\Theta} - \eta \cdot \nabla_{\boldsymbol{\Theta}} \mathcal{L}_{\mathrm{rec}}, \qquad (14)$$

where $\eta$ denotes the adaptation learning rate and we denote all components of the world model without policies collectively by parameters $\boldsymbol{\Theta}$.

This online update allows SC$^2$-WM to refine its internal dynamics in testing scenarios, working synergistically with plan refinement to enable effective navigation. The complete procedure of the proposed conditional world-aware adaptation is summarized in Algorithm 2.

---

**Algorithm 2** Conditional World-Aware Adaptation *(Test-Time)*

---

**Input:** Pre-trained world model parameters $\Theta$, adaptation threshold $\tau$, learning rate $\eta$
**Output:** Adapted world model parameters $\Theta'$
1: **for** each testing step $t$ **do**
2:     *// Compute action distributions*
3:     $\tilde{\boldsymbol{p}}_t = \tilde{\pi}(\boldsymbol{s}_t)$
4:     $\boldsymbol{p}_t = \pi(\boldsymbol{s}_t^{(\Delta)})$
5:     *// Measure feedback-induced discrepancy*
6:     $\kappa_t = D_{\mathrm{KL}}(\tilde{\boldsymbol{p}}_t \,\|\, \boldsymbol{p}_t)$
7:     *// Conditional adaptation trigger*
8:     **if** $\kappa_t > \tau$ **then**
9:         $\mathcal{L}_{\mathrm{rec}} = \|d(\tilde{\boldsymbol{s}}_{t+1}) - \hat{\boldsymbol{v}}_t\|_2^2$
10:       $\Theta \leftarrow \Theta - \eta \cdot \nabla_{\Theta} \mathcal{L}_{\mathrm{rec}}$
11:     **end if**
12: **end for**

---

## 4. Experiments

To validate the effectiveness of our proposed method, we conduct experiments on two continuous environment benchmarks: R2R-CE and RxR-CE (Krantz et al., 2020), which are continuous reconstructions of the discrete R2R (Anderson et al., 2018b) and RxR (Ku et al., 2020) datasets. Additional experimental results and ablation studies are provided in Sec. D of the Appendix.

### 4.1. Experimental Setup

**Datasets.** The R2R-CE dataset (Krantz et al., 2020; Anderson et al., 2018b) consists of 5,611 shortest-path trajectories distributed across training, validation, and test sets. Each path is associated with an average of three English instructions. The trajectories feature a mean length of 9.89 meters, while the instructions have an average length of 32 words.The RxR-CE dataset (Krantz et al., 2020; Ku et al., 2020), which shares similar scene splits, presents a more challenging setting with a larger scale and multilingual instructions (English, Hindi, and Telugu). The instructions here are significantly longer, averaging 120 words.A key distinction lies in the agent dynamics: R2R-CE agents possess a chassis radius of 0.10 meters and are permitted to slide along obstacles. In contrast, RxR-CE agents have a larger radius of 0.18 meters and are strictly prohibited from sliding, making them more susceptible to collisions.

**Evaluation Metrics.** We employ a comprehensive set of metrics used in prior works(Anderson et al., 2018a;b; Ilharco et al., 2019): Trajectory Length (TL), Navigation Error (NE), Success Rate given Oracle stop policy (OSR), Success Rate (SR), Success weighted by Path Length (SPL), Normalized Dynamic Time Warping (NDTW), and Success weighted by NDTW (SDTW). More details are provided in

*Table 1.* Experimental results on R2R-CE dataset. Results better than **base model** are shown in blue. Best results for both panoramic and monocular settings are underlined. * indicates experimental results that we have reproduced.

| Camera | Methods | Val Seen | | | | | Val Unseen | | | | | Test Unseen | | | | |
|---|---|---|---|---|---|---|---|---|---|---|---|---|---|---|---|---|
| | | TL ↓ | NE ↓ | OSR | SR | SPL | TL ↓ | NE ↓ | OSR | SR | SPL | TL ↓ | NE ↓ | OSR | SR | SPL |
| **Monocular** | LAW (Raychaudhuri et al., 2021) [EMNLP21] | 9.34 | 6.35 | 49 | 40 | 37 | 8.89 | 6.83 | 44 | 35 | 31 | 9.67 | 7.69 | 28 | 38 | 25 |
| | CM² (Georgakis et al., 2022) [CVPR22] | 12.05 | 6.10 | 50.7 | 42.9 | 34.8 | 11.54 | 7.02 | 41.5 | 34.3 | 27.6 | 13.90 | 7.70 | 39 | 31 | 24 |
| | WS-MGMap (Chen et al., 2022a) [NeurIPS22] | 10.12 | 5.65 | 51.7 | 46.9 | 43.4 | 10.00 | 6.28 | 47.6 | 38.9 | 34.3 | 12.30 | 7.11 | 45 | 35 | 28 |
| | NaVid (Zhang et al., 2024) [RSS24] | - | - | - | - | - | - | 5.47 | 49.1 | 37.4 | 35.9 | - | - | - | - | - |
| | ETPNav/p (Wang et al., 2025c) [CoRL24] | - | - | - | - | - | - | 6.81 | 42.4 | 32.9 | 23.1 | - | - | - | - | - |
| | NavMorph (Yao et al., 2025) [ICCV2025] | 20.03 | 4.58 | 62.7 | 55.8 | 38.9 | 22.54 | 5.75 | 56.9 | 47.9 | 33.2 | 24.75 | 6.01 | 54.5 | 45.7 | 30.2 |
| | g3D-LF (Wang & Lee, 2025) [CVPR2025] | - | - | - | - | - | - | 5.70 | 59.5 | 47.2 | 34.6 | - | 6.00 | 57.5 | 46.3 | 32.2 |
| | VLN-3DFF (Wang et al., 2025c) [CoRL24] | - | - | - | - | - | - | 5.95 | 55.8 | 44.9 | 30.4 | - | 6.24 | 54.4 | 43.7 | 28.9 |
| | VLN-3DFF* | 22.90 | 4.92 | 62.1 | 52.7 | 36.7 | 26.16 | 6.05 | 54.9 | 43.8 | 29.4 | 26.02 | 6.22 | 54.7 | 43.8 | 28.6 |
| | **SC²-WM** | 17.26 | 4.53 | 64.3 | 56.0 | 41.9 | 17.65 | 5.37 | 58.8 | 50.9 | 37.2 | 21.68 | 6.04 | 57.1 | 47.0 | 32.1 |
| **Panoramic** | Seq2Seq (Anderson et al., 2018b) [CVPR18] | 9.26 | 7.12 | 46 | 37 | 35 | 8.64 | 7.37 | 40 | 32 | 30 | 8.85 | 7.91 | 36 | 28 | 25 |
| | Sim2Sim (Krantz & Lee, 2022) [ECCV22] | 11.18 | 4.67 | 61 | 52 | 44 | 10.69 | 6.07 | 52 | 43 | 36 | 11.43 | 6.17 | 52 | 44 | 37 |
| | CWP-CMA (Hong et al., 2022) [CVPR22] | 11.47 | 5.20 | 61 | 51 | 45 | 10.90 | 6.20 | 52 | 41 | 36 | 11.85 | 6.30 | 49 | 38 | 33 |
| | CWP-BERT (Hong et al., 2022) [CVPR22] | 12.50 | 5.02 | 59 | 50 | 44 | 12.23 | 5.74 | 53 | 44 | 39 | 13.51 | 5.89 | 51 | 42 | 36 |
| | DREAMW (Wang et al., 2023a) [ICCV23] | 11.60 | 4.09 | 59 | 66 | 48 | 11.30 | 5.53 | 49 | 59 | 44 | 11.80 | 5.48 | 49 | 57 | 44 |
| | GridMM (Wang et al., 2023c) [ICCV23] | 12.69 | 4.21 | 69 | 59 | 51 | 13.36 | 5.11 | 61 | 49 | 41 | 13.31 | 5.64 | 56 | 46 | 39 |
| | BEVBert (An et al., 2022a) [ICCV23] | 13.98 | 3.77 | 73 | 68 | 60 | 13.27 | 4.57 | 67 | 59 | 50 | 15.31 | 4.70 | 67 | 59 | 50 |
| | FSTTA (Gao et al., 2024) [ICML24] | 12.39 | 4.25 | 69 | 58 | 50 | 11.58 | 5.27 | 58 | 48 | 42 | 13.17 | 5.84 | 55 | 46 | 38 |
| | Dynam3D (Wang et al., 2025b) [ICCV2025] | - | - | - | - | - | - | 5.34 | 62 | 53 | 46 | - | 5.53 | 60 | 51 | 45 |
| | NavMorph (Yao et al., 2025) [ICCV2025] | 11.76 | 3.66 | 78 | 70 | 62 | 12.68 | 4.37 | 68 | 64 | 53 | 12.69 | 4.69 | 68 | 60 | 52 |
| | g3D-LF (Wang & Lee, 2025) [CVPR2025] | - | - | - | - | - | - | 4.53 | 68 | 61 | 52 | - | 4.78 | 68 | 58 | 51 |
| | ETPNav (An et al., 2024) [TPAMI24] | 11.78 | 3.95 | 72 | 66 | 59 | 11.99 | 4.71 | 65 | 57 | 49 | 12.87 | 5.12 | 63 | 55 | 48 |
| | ETPNav* | 11.35 | 3.93 | 72 | 66 | 59 | 11.40 | 4.69 | 64 | 57 | 49 | 12.72 | 5.10 | 63 | 55 | 48 |
| | **SC²-WM** | 12.15 | 3.85 | 73 | 68 | 60 | 12.05 | 4.60 | 68 | 59 | 51 | 13.88 | 4.97 | 65 | 57 | 50 |
| | HNR (Wang et al., 2024b) [CVPR24] | 11.79 | 3.67 | 76 | 69 | 61 | 12.64 | 4.42 | 67 | 61 | 51 | 13.03 | 4.81 | 67 | 58 | 50 |
| | HNR* | 11.84 | 3.73 | 76 | 69 | 61 | 12.76 | 4.57 | 67 | 61 | 51 | 12.92 | 4.85 | 67 | 58 | 50 |
| | **SC²-WM** | 12.09 | 3.28 | 80 | 71 | 64 | 12.89 | 4.25 | 70 | 66 | 54 | 13.42 | 4.90 | 71 | 62 | 53 |

Note: Following prior work, we report the results with different precision formats across camera configurations——integers for panoramic settings and two decimal places for monocular settings.

Sec. B of the Appendix.

**Implementation Details.** We evaluate our method in both monocular and panoramic navigation settings. In the monocular setting, we adopt the VLN-3DFF framework (Wang et al., 2025c), which leverages the pretrained 3D Feature Fields (Wang et al., 2024b) to enable monocular agents to operate in environments originally designed for panoramic observations. This setting better reflects practical deployment constraints, where monocular cameras provide advantages in cost and energy efficiency.

Under the panoramic setting, we follow the standard VLN-CE protocol (Krantz et al., 2020; 2021), where 12 RGB-D observations are captured at $30°$ intervals at each location. We integrate SC²-WM with ETPNav (An et al., 2024) and HNR (Wang et al., 2024b) to verify its compatibility and generalizability across different navigation architectures.

To perform conditional adaptation, we maintain a dynamic percentile-based queue over recent discrepancy values $\kappa_t$, and trigger adaptation when the current discrepancy exceeds the $\tau$-th percentile ($\tau = 0.5$). The refinement function $\psi$ in Eq. (10) is implemented as a lightweight dynamic gating network that predicts residual corrections conditioned on reconstructed state and action features. A sigmoid gate is further employed to adaptively modulate the update magnitude. The entire framework is trained end-to-end using the objectives described in Sec. 3.2.

All experiments follow the online VLN evaluation protocol (Gao et al., 2024) with batch size 1 during inference. Our implementation is based on PyTorch and trained on a single NVIDIA RTX 3090 GPU. Additional implementation details are provided in Sec. C of the Appendix.

### 4.2. Comparison with State-of-the-art VLN Models

**R2R-CE.** Table 1 presents a comprehensive comparison between our proposed SC²-WM and previous state-of-the-art methods on R2R-CE datasets. Our method significantly outperforms the strong baseline, VLN-3DFF, across multiple metrics. On the Val Unseen split, we achieve remarkable gains of 7.1% in Success Rate (SR) and nearly 7.8% in Success weighted by Path Length (SPL). Simultaneously, the method demonstrates a sharp reduction in Trajectory Length (TL), with an average decrease of 5 meters. These gains generalize robustly to the Test Unseen split, where SC²-WM surpasses baseline by 3.2% in SR and 3.5% in SPL. Compared with the recent method g3D-LF, our approach achieves superior or comparable performance on most metrics. It is worth noting that g3D-LF leverages additional 3D representations, whereas our method does not. Besides, recent monoVLN method (Lu et al., 2025) employs more powerful 3DGS-based feature fields and achieves superior performance. Since monoVLN has not been open-sourced, we are unable to adopt it as a base model for world model construction. Incorporating more expressive 3D representa-

*Table 2.* Experimental results on RxR-CE dataset. Results better than **base model** are shown in blue. Best results for both panoramic and monocular settings are underlined. * indicates experimental results that we have reproduced.

| Camera | Methods | Val Unseen | | | | |
|---|---|---|---|---|---|---|
| | | NE ↓ | SR | SPL | NDTW | SDTW |
| Monocular | LAW (Raychaudhuri et al., 2021) | 10.87 | 8.0 | 8.0 | - | - |
| | CM² (Georgakis et al., 2022) | 8.98 | 14.4 | 9.2 | - | - |
| | WS-MGMap (Chen et al., 2022a) | 9.83 | 15.0 | 12.1 | - | - |
| | NaVid (Zhang et al., 2024) | 8.41 | 23.8 | 32.2 | - | - |
| | A²-Nav (Chen et al., 2023) | - | 16.8 | 6.3 | - | - |
| | NavMorph (Yao et al., 2025) | 8.85 | 30.8 | 22.8 | 44.2 | 23.3 |
| | VLN-3DFF (Wang et al., 2025c) | 8.79 | 25.5 | 18.1 | - | - |
| | VLN-3DFF* | 9.40 | 26.7 | 20.1 | 42.9 | 20.4 |
| | **SC²-WM** | 8.36 | 35.8 | 27.2 | 44.9 | 26.5 |
| Panoramic | LAW-Pano (Raychaudhuri et al., 2021) | 11.04 | 10 | 9 | - | - |
| | Seq2Seq (Anderson et al., 2018b) | 12.10 | 14 | 12 | 31 | 11 |
| | CWP-CMA (Hong et al., 2022) | 8.76 | 27 | 22 | 47 | - |
| | CWP-BERT (Hong et al., 2022) | 8.98 | 27 | 23 | 47 | - |
| | AO-Planner (Chen et al., 2024) | 7.06 | 43 | 30 | 50 | - |
| | Reborn (An et al., 2022b) | 5.98 | 49 | 42 | 63 | 42 |
| | NavMorph (Yao et al., 2025) | 5.70 | 58 | 49 | 65 | 49 |
| | ETPNav (An et al., 2024) | 5.64 | 55 | 45 | 62 | 45 |
| | ETPNav* | 5.96 | 55 | 45 | 61 | 45 |
| | **SC²-WM** | 5.57 | 57 | 46 | 64 | 47 |
| | HNR (Wang et al., 2024b) | 5.51 | 56 | 47 | 64 | 47 |
| | HNR* | 5.75 | 56 | 47 | 63 | 47 |
| | **SC²-WM** | 5.72 | 60 | 50 | 67 | 50 |

Note: Following prior work, we report the results with different precision formats across camera configurations, integers for panoramic settings and two decimal places for monocular settings.

tions will be explored as future work.

We attribute the reduction in trajectory length directly to our correction mechanism. By anticipating potential future observations before executing an action, the agent effectively prunes useless exploration and avoids sub-optimal steps. Regarding the improvements in SR and SPL, these stem from the intrinsic nature of the predictive module, alongside the effective utilization of historical memory. The objective of successfully predicting future visual features implicitly compels the model to develop a deeper cognitive understanding of the environment; this enhanced internal representation indirectly strengthens the agent's overall decision-making capability. Furthermore, our online Test-Time Adaptation (TTA) plays a critical role in generalization, as it continuously refines the World Model's predictive accuracy to maintain high environmental awareness even in unfamiliar, unseen settings. In the panoramic setting, SC²-WM achieves the best overall results on the Val Unseen split, outperforming HNR by +5% SR and +3% SPL, indicating more stable and goal-aligned decision making.

**RxR-CE.** Table 2 extends our evaluation to the RxR-CE dataset, where SC²-WM achieves comprehensive improvements over VLN-3DFF. We observe substantial gains of 9.1% in SR and 7.1% in SPL, alongside a consistent reduction in average trajectory length ($8.36\,m$ vs $9.41\,m$). Considering the strict sequentiality and temporal complexity of RxR instructions, these results underscore the necessity of our correction mechanism, which ensures the agent adheres closer to the described path, yielding higher path fidelity as evidenced by the significant boost in SDTW (26.5% vs 20.4%). As for panoramic setting, when built upon ETPNav,

*Table 3.* Ablation Study of the proposed World Model. Best results are highlighted in **bold**.

| Model | State-C | VCM | Model-C | R2R-CE Val Unseen | | | | | | |
|---|---|---|---|---|---|---|---|---|---|---|
| | | | | TL↓ | NE↓ | OSR | SR | SPL | NDTW | SDTW |
| Base model | - | - | - | 26.16 | 6.05 | 54.92 | 43.77 | 29.39 | 40.94 | 29.30 |
| SC²-WM | ✓ | | | 21.75 | 5.69 | 56.22 | 48.34 | 33.02 | 45.36 | 32.82 |
| | ✓ | ✓ | | 18.73 | 5.46 | **59.22** | 49.86 | 36.05 | 48.10 | 34.62 |
| | ✓ | ✓ | ✓† | 18.43 | 5.43 | 58.67 | 50.30 | 36.58 | 48.66 | 35.37 |
| | ✓ | ✓ | ✓ | **17.65** | **5.37** | 58.84 | **50.90** | **37.17** | **49.31** | **35.70** |
| SC²-WM *w/o* $\mathcal{L}_{rec}$ | ✓ | ✓ | ✓ | 18.12 | 5.55 | 57.37 | 48.89 | 34.58 | 47.62 | 33.91 |

Note: State-C and Model-C denote State-Level Correction and Model-Level Correction, respectively. † indicates fixed-interval adaptation performed every $T = 2$ steps, instead of the proposed feedback-triggered adaptation strategy.

our method improves SR from 55% to 57% and SPL from 45% to 47% on the val-unseen split, while reducing NE from $5.96\,m$ to $5.57\,m$, indicating more efficient navigation paths. Notably, the consistent improvements across both frameworks demonstrate that our self-correcting mechanism is complementary to existing approaches and effectively mitigates error accumulation through closed-loop feedback.

**Discussion.** While recent MLLM-based VLN approaches (full comparisons are provided in Table 4 of the Appendix) primarily improve navigation through scaling multimodal semantic reasoning, our work focuses on a lightweight world-model paradigm that enhances execution-time adaptability and decision stability within a compact framework. Notably, SC²-WM requires only a single NVIDIA RTX 3090 GPU and less than 40 hours of training, demonstrating that effective closed-loop self-correction can be achieved without relying on large-scale multimodal models. Further discussion is provided in Sec. D.2 of the Appendix.

### 4.3. Further Remarks

**Ablation Study of the Proposed World Model.** To investigate the efficacy of each component in SC²-WM, we conduct ablation studies by incrementally incorporating the feedback-guided plan refinement ('State-C'), the visual calibration module ('VCM'), and the conditional world-aware adaptation ('Model-C'). Here, state-level correction refers to feedback-guided plan refinement that adjusts latent representations before action execution, while model-level correction refers to conditional world-aware adaptation that selectively updates the world model at test time. Table 3 details the contribution of each component on the R2R-CE Val Unseen split:

- **Effect of State-Level Self-Correction.** Introducing feedback-guided plan refinement leads to a direct increase in SR from 43.77% to 48.34%, alongside a reduction in TL (from $26.16\,m$ to $21.75\,m$). This indicates that the state-level correction effectively identifies and prunes erroneous steps by adjusting latent states based on world-model foresight.

- **Impact of VCM.** The integration of the VCM module triggers a substantial performance leap, reducing TL to $18.73\,m$ and boosting SPL to 36.05%. Episodic

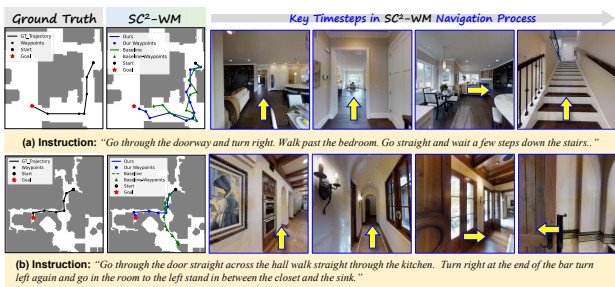

*Figure 3.* **Qualitative results on R2R-CE unseen set.** Comparison between our SC²-WM (blue) and baseline VLN-3DFF (green) against ground truth (black). Yellow arrows indicate the agent's moving direction.

Memory in VCM provides the world model with essential historical context, enriching its foresight capability and enabling more accurate feedback signals for plan refinement.

• **Benefit of Model-Level Correction.** Conditional world-aware adaptation further elevates performance by selectively updating the world model online. By adapting the predictive module to unseen environments, model-level correction achieves the best overall performance with an SR of 50.9% and the lowest TL of $17.65\,m$, validating the importance of dynamic adaptation for generalization. Notably, compared to the fixed-interval update mechanism adopted in (Gao et al., 2024), where we set $T = 2$ to match the average update frequency of our method for fair comparison, our feedback-triggered approach yields higher SR and SPL, demonstrating that selective adaptation based on internal feedback signals outperforms naive periodic updates. Additionally, removing the reconstruction loss $\mathcal{L}_{\text{rec}}$ (Eq. (13)) leads to performance degradation, confirming that $\mathcal{L}_{\text{rec}}$ helps learn effective latent representations for accurate world model predictions.

**Qualitative Analysis.** Figure 3 presents qualitative comparisons on the VLN-CE task using the R2R-CE dataset. We visualize two navigation episodes with varying instruction complexity. In both cases, our SC²-WM (blue) produces trajectories that closely align with the ground truth (black), while the baseline method (green) tends to deviate from the intended path after a few steps. Notably, in Figure 3(b) where the instruction involves multiple turns and spatial references, our method maintains correct navigation throughout the episode, whereas the baseline fails to recover after early mistakes. These results suggest that our dual world-model design enables better spatial reasoning and more robust long-horizon navigation.

**Real-world Experiments.** We deploy SC²-WM on a physical robotic platform to validate its effectiveness beyond sim-

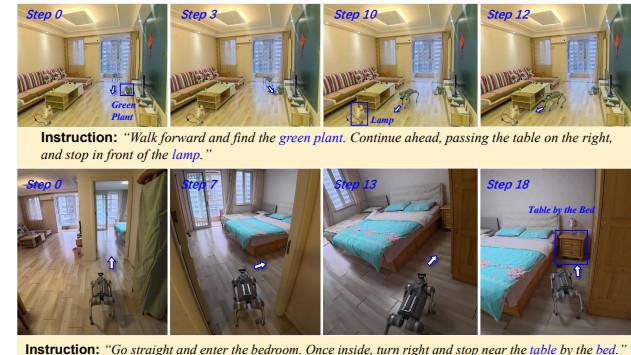

*Figure 4.* **Real-world deployment on a Unitree GO2 quadruped robot.** SC²-WM enables robust vision-and-language navigation in indoor environments, leveraging internal feedback for self-correction under complex instructions.

ulation. The platform consists of a Unitree GO2 quadruped robot equipped with an Intel RealSense D435i RGB-D camera for visual perception. Given natural language navigation instructions, the robot is required to navigate through indoor environment. As shown in Figure 4, SC²-WM demonstrates robust navigation behavior across varying conditions, highlighting the benefit of closed-loop self-correction for real-world deployment. Details are provided in the Appendix.

## 5. Conclusion

We presented SC²-WM, a self-correcting world model framework that enables closed-loop decision making for vision-and-language navigation. By deriving internal feedback from world-model foresight, our approach performs state-level plan refinement and model-level adaptation, providing a principled way to mitigate error accumulation under partial observability.

The current framework opens several promising directions for future research. The quality of internal feedback is tied to world-model expressiveness, suggesting potential benefits from incorporating uncertainty-aware or structured world models. Additionally, the test-time adaptation mechanism could be further optimized for efficiency to enable deployment on resource-constrained platforms. We also plan to extend the framework to longer-horizon planning and multi-agent scenarios, and explore tighter integration between language grounding and world modeling for richer semantic feedback.

## Acknowledgments

This work was supported in part by the Guangdong S&T Program (2024B0101050004), in part by the National Natural Science Foundation of China under Grants 62472422, U23A20387, and 62236008.

## Impact Statement

This paper presents work whose goal is to advance the field of Machine Learning. There are many potential societal consequences of our work, none which we feel must be specifically highlighted here.

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

§ A presents the guidelines for our source code. § B presents more details about evaluation metrics. Implementation details for experiments are provided in § C, and further comparisons against state-of-the-art methods are shown in § D. Finally, we present real-world verification to validate the effectiveness of our method in § E.

## A. Source Code

The source code to reproduce our experimental results is included in the supplementary material (6618-supp.zip) under the `./code` directory. For more details, please refer to the README file therein.

## B. Evaluation Metrics for VLN-CE agents

We follow previous approaches (Anderson et al., 2018b;a; Ilharco et al., 2019) and adopt the standard metrics for evaluating VLN-CE agents:

- **TL (Trajectory length)** measures the average length of the predicted navigation trajectories.

- **NE (Navigation Error)** measures the average distance (in meter) between the agent's final position in the predicted trajectory and the target in the ground truth.

- **SR (Success Rate)** is the proportion of the agent stopping in the predicted route within a threshold distance (set as 3 meters) of the goal in the reference route.

- **OSR (Oracle Success Rate)** is the proportion of the closest point in the predicted trajectory to the target in the reference trajectory within a threshold distance.

- **SPL (Success weighted by Path Length)** is a comprehensive metric method integrating SR and TL that takes both effectiveness and efficiency into account.

- **NDTW (Normalized Dynamic Time Warping)** measures the normalized cumulative distance between reference path and agent position.

- **SDTW (Success weighted by normalized Dynamic Time Warping)** is a comprehensive metric method integrating NDTW and SR that takes both path efficiency and task completion into account.

## C. Implementation Details

**The Baseline Framework.** In conventional panoramic VLN-CE frameworks (An et al., 2024; Wang et al., 2023c), the agent perceives its surroundings through multi-view RGB-D panoramas captured at 30-degree intervals at each timestep $t$. These panoramic observations are processed by a trained waypoint prediction module(Hong et al., 2022) to identify navigable waypoints. The VLN model then encodes both the visual features of these waypoints and their spatial information (relative direction and distance) to construct a topological map. This map is subsequently integrated with the navigation instruction via the Cross-Modal Graph Transformer(Chen et al., 2022c; An et al., 2024), which selects the optimal waypoint as the agent's next navigation goal.

Monocular VLN-CE settings rely on a single RGB-D camera, which presents challenges in waypoint estimation due to the lack of full panoramic coverage. To address this, an enhanced waypoint predictor (Wang et al., 2025c) utilizes a semantic traversability map and 3D feature fields to infer viable waypoints, ensuring effective decision-making even with limited field-of-view.

**Model Configuration.** Following previous baseline models (An et al., 2024; Wang et al., 2025c), we utilize CLIP-pretrained ViT-B/16 (Dosovitskiy, 2020) for RGB feature extraction on R2R-CE, while ViT-B/32 (Dosovitskiy, 2020) is adopted for RxR-CE to maintain consistency with prior settings. Depth information is processed using a point-goal navigation pretrained ResNet-50 (He et al., 2016). The framework maintains encoder depths of 2, 9, and 4 layers for panoramic, textual, and cross-modal graph components respectively, aligned with (Hong et al., 2022; Georgakis et al., 2022). Other hyperparameters are the same as LXMERT (Tan & Bansal, 2019) on the R2R-CE dataset and pre-trained RoBerta (Liu, 2019) for the multilingual RxR-CE dataset. The camera's HFOV is set to $90°$ for R2R-CE and $79°$ for RxR-CE.

**Experimental Details.** We conduct our experiments using distinct initialization and training strategies for the R2R-CE and RxR-CE datasets. For R2R-CE, we initialize the model with weights pretrained on the R2R-CE dataset for 25,000 iterations. We employ a hierarchical training strategy consisting of two stages: First, we jointly train all modules with a learning rate of $1 \times 10^{-5}$ for 12,000 episodes. Subsequently, we freeze the language encoder and fine-tune the remaining components with a reduced learning rate of $4 \times 10^{-6}$ for an additional 10,000 episodes. For RxR-CE, we initialize our framework using the pretrained checkpoint from VLN-3DFF (Wang et al., 2025c). The model is then trained with a learning rate of $1 \times 10^{-5}$ for 14,000 episodes.

In addition, We use a dynamic percentile-based queue over recent , triggering adaptation when it exceeds the $\tau$-th percentile ($\tau = 0.5$). This self-adapts to recent dynamics without manual threshold tuning. $\psi$ in Eq. (10) is implemented as a dynamic gating network that predicts a residual correction conditioned on reconstructed state and action features. A sigmoid gate modulates the update magnitude adaptively. The network is trained end-to-end with the same objectives.

**Temporal Prior in Memory of VCM module.** To explicitly incorporate temporal locality and prioritize recent navigational contexts, we inject a learnable temporal bias into the cross-attention mechanism of the memory $\mathcal{M}_t$. We introduce a learnable scalar parameter $\lambda$ (initialized to 0.1) to regulate the attention distribution based on the temporal distance between the current time step $t$ and a historical memory step $k$. A bias term, formulated as $B_{t,k} = -|\lambda| \cdot (t - k)$, is added to the raw attention logits prior to the softmax normalization. This linear decay effectively penalizes older historical observations, encouraging the agent to assign higher attention weights to the most immediate predecessor steps during the reasoning process.

**Spatio-temporal Encodings in Memory of VCM Module.** To effectively ground historical observations within the agent's current frame of reference, we enhance the visual features of memory items with explicit spatial and temporal encodings prior to the attention mechanism:

**(i) Relative Spatial Encoding.** We compute a 7-dimensional relative geometric feature vector $\mathbf{g}_{t,k} \in \mathbb{R}^7$ for each historical node $k$ with respect to the agent's current pose at step $t$. This vector consists of the relative heading and elevation (represented by sine and cosine values) and the normalized Euclidean distance. These geometric features are projected into the model's hidden dimension $D$ via a linear layer followed by Layer Normalization:

$$\mathbf{e}_{t,k}^{(pos)} = \text{LayerNorm}(\mathbf{W}_p \mathbf{g}_{t,k} + \mathbf{b}_p),\tag{15}$$

where $\mathbf{W}_p$ and $\mathbf{b}_p$ are learnable parameters.

**(ii) Step (Temporal) Embedding.** To encode the sequential order and temporal distance of visited nodes, we introduce a discrete step embedding. Let $\delta t = \min(t - k, T_{max})$ denote the clamped time difference between the current step $t$ and the memory step $k$. We utilize a learnable lookup table $\mathbf{E}_{step}$ to retrieve a dense vector representation for this time lag:

$$\mathbf{e}_{t,k}^{(step)} = \mathbf{E}_{step}[\delta t].\tag{16}$$

Both $\mathbf{e}_{t,k}^{(pos)}$ and $\mathbf{e}_{t,k}^{(step)}$ are added element-wise to the visual features of the corresponding memory item, thereby fusing spatiotemporal context into the visual representation.

# D. Complementary Experiments

## D.1. Full Results

In our main paper, we provide representative comparison results on the R2R-CE (Krantz et al., 2020) and RxR-CE (Krantz et al., 2020) benchmarks due to space constraints. Here, we present the complete results across the 'validation seen', 'validation unseen', and 'test unseen' splits of these benchmarks, including comparisons with a broader range of state-of-the-art methods, as detailed in Table 4 and Table 5. Our proposed SC$^2$-WM method enhances its ability to perform navigation decision based on self-correcting with closed-loop feedback, effectively handling complex navigation tasks even with monocular input. Note that the recent monoVLN method (Lu et al., 2025) employs more powerful 3DGS-based feature fields and achieves superior performance. Since monoVLN has not been open-sourced, we are unable to adopt it as a base model for world model construction. Incorporating more expressive 3D representations will be explored as future work. Besides, compared with monoVLN, our method achieves better performance on the RxR dataset.

## D.2. Discussion on Model Paradigm

While we include MLLM-based VLN methods in the full results (Table 4), our work focuses on a lightweight world-model-based framework rather than incorporating large-scale multimodal language models. Although recent MLLM-based approaches demonstrate strong semantic reasoning, they typically incur substantial computational overhead and remain limited in online adaptability during execution. In contrast, prior studies have shown that improving internal state representations and predictive dynamics can effectively enhance decision stability under partial observability (Yao et al., 2025; Wang & Lee, 2025; Huang et al., 2025; An et al., 2024). Building on this line of work, our design introduces internal feedback and self-correction within a compact world model, enabling robust execution-time regulation without relying on large-scale models. It is worth noting that the model proposed in this paper requires only a single NVIDIA RTX 3090 GPU and less than 40 hours of training to achieve strong performance, whereas such computational resources are typically insufficient to meet the demands of methods based on large-scale multimodal models.

*Table 4.* Experimental results on R2R-CE dataset. Results better than **base model** are shown in blue. Best results for **both panoramic and monocular settings** are underlined. * indicates experimental results that we have reproduced. [†] Methods based on large language/vision-language models.

| Camera | Methods | Val Seen | | | | | Val Unseen | | | | | Test Unseen | | | | |
|---|---|---|---|---|---|---|---|---|---|---|---|---|---|---|---|---|
| | | TL ↓ | NE ↓ | OSR | SR | SPL | TL ↓ | NE ↓ | OSR | SR | SPL | TL ↓ | NE ↓ | OSR | SR | SPL |
| Monocular | LAW (Raychaudhuri et al., 2021) [EMNLP21] | 9.34 | 6.35 | 49 | 40 | 37 | 8.89 | 6.83 | 44 | 35 | 31 | 9.67 | 7.69 | 28 | 38 | 25 |
| | CM² (Georgakis et al., 2022) [CVPR22] | 12.05 | 6.10 | 50.7 | 42.9 | 34.8 | 11.54 | 7.02 | 41.5 | 34.3 | 27.6 | 13.90 | 7.70 | 39 | 31 | 24 |
| | WS-MGMap (Chen et al., 2022a) [NeurIPS22] | 10.12 | 5.65 | 51.7 | 46.9 | 43.4 | 10.00 | 6.28 | 47.6 | 38.9 | 34.3 | 12.30 | 7.11 | 45 | 35 | 28 |
| | NaVid (Zhang et al., 2024) [RSS24] | - | - | - | - | - | - | 5.47 | 49.1 | 37.4 | 35.9 | - | - | - | - | - |
| | ETPNav/p (Wang et al., 2025c) [CoRL24] | - | - | - | - | - | - | 6.81 | 42.4 | 32.9 | 23.1 | - | - | - | - | - |
| | NavMorph (Yao et al., 2025) [ICCV2025] | 20.03 | 4.58 | 62.7 | 55.8 | 38.9 | 22.54 | 5.75 | 56.9 | 47.9 | 33.2 | 24.75 | 6.01 | 54.5 | 45.7 | 30.2 |
| | g3D-LF (Wang & Lee, 2025) [CVPR2025] | - | - | - | - | - | - | 5.70 | 59.5 | 47.2 | 34.6 | - | 6.00 | 57.5 | 46.3 | 32.2 |
| | monoVLN (Lu et al., 2025) [CVPR2025] | - | - | - | - | - | - | 4.61 | 62.4 | 54.8 | 44.4 | - | 4.97 | 60.5 | 53.6 | 44.9 |
| | NaVILA[†] (Cheng et al., 2024) [arxiv2025] | - | - | - | - | - | - | 5.43 | 62.5 | 54.0 | 49.0 | - | - | - | - | - |
| | UniNaVid[†] (Zhang et al., 2024) [arxiv2025] | - | - | - | - | - | - | 5.58 | 53.3 | 47.0 | 42.7 | - | - | - | - | - |
| | StreamVLN[†] (Wei et al., 2025b) [arxiv2025] | - | - | - | - | - | - | 4.98 | 64.2 | 56.9 | 51.9 | - | - | - | - | - |
| | DualVLN[†] (Wei et al., 2025a) [arxiv2025] | - | - | - | - | - | - | 4.05 | 70.7 | 64.3 | 58.5 | - | - | - | - | - |
| | VLN-3DFF (Wang et al., 2025c) [CoRL24] | - | - | - | - | - | - | 5.95 | 55.8 | 44.9 | 30.4 | - | 6.24 | 54.4 | 43.7 | 28.9 |
| | VLN-3DFF* | 22.90 | 4.92 | 62.1 | 52.7 | 36.7 | 26.16 | 6.05 | 54.9 | 43.8 | 29.4 | 26.02 | 6.22 | 54.7 | 43.8 | 28.6 |
| | **SC²-WM** | 17.26 | 4.53 | 64.3 | 56.0 | 41.9 | 17.65 | 5.37 | 58.8 | 50.9 | 37.2 | 21.68 | 6.04 | 57.1 | 47.0 | 32.1 |
| Panoramic | Seq2Seq (Anderson et al., 2018b) [CVPR18] | 9.26 | 7.12 | 46 | 37 | 35 | 8.64 | 7.37 | 40 | 32 | 30 | 8.85 | 7.91 | 36 | 28 | 25 |
| | Sim2Sim (Krantz & Lee, 2022) [ECCV22] | 11.18 | 4.67 | 61 | 52 | 44 | 10.69 | 6.07 | 52 | 43 | 36 | 11.43 | 6.17 | 52 | 44 | 37 |
| | CWP-CMA (Hong et al., 2022) [CVPR22] | 11.47 | 5.20 | 61 | 51 | 45 | 10.90 | 6.20 | 52 | 41 | 36 | 11.85 | 6.30 | 49 | 38 | 33 |
| | CWP-BERT (Hong et al., 2022) [CVPR22] | 12.50 | 5.02 | 59 | 50 | 44 | 12.23 | 5.74 | 53 | 44 | 39 | 13.51 | 5.89 | 51 | 42 | 36 |
| | DREAMW (Wang et al., 2023a) [ICCV23] | 11.60 | 4.09 | 59 | 66 | 48 | 11.30 | 5.53 | 49 | 59 | 44 | 11.80 | 5.48 | 49 | 57 | 44 |
| | GridMM (Wang et al., 2023c) [ICCV23] | 12.69 | 4.21 | 69 | 59 | 51 | 13.36 | 5.11 | 61 | 49 | 41 | 13.31 | 5.64 | 56 | 46 | 39 |
| | BEVBert (An et al., 2022a) [ICCV23] | 13.98 | 3.77 | 73 | 68 | 60 | 13.27 | 4.57 | 67 | 59 | 50 | 15.31 | 4.70 | 67 | 59 | 50 |
| | InstructNav[†] (Long et al., 2024) [arxiv2024] | - | - | - | - | - | 7.74 | 6.89 | - | 31 | 24 | - | - | - | - | - |
| | FSTTA (Gao et al., 2024) [ICML24] | 12.39 | 4.25 | 69 | 58 | 50 | 11.58 | 5.27 | 58 | 48 | 42 | 13.17 | 5.84 | 55 | 46 | 38 |
| | SmartWay[†] (Shi et al., 2025) [arxiv2025] | - | - | - | - | - | 13.09 | 7.01 | 51 | 29 | 22 | - | - | - | - | - |
| | Aux-Think[†] (Wang et al., 2025a) [arxiv2025] | - | - | - | - | - | - | 6.01 | 52 | 46 | 41 | - | - | - | - | - |
| | Dynam3D[†] (Wang et al., 2025b) [ICCV2025] | - | - | - | - | - | - | 5.34 | 62 | 53 | 46 | - | 5.53 | 60 | 51 | 45 |
| | NavMorph (Yao et al., 2025) [ICCV2025] | 11.76 | 3.66 | 78 | 70 | 62 | 12.68 | 4.37 | 66 | 54 | 53 | 12.69 | 4.69 | 68 | 60 | 52 |
| | g3D-LF (Wang & Lee, 2025) [CVPR2025] | - | - | - | - | - | - | 4.53 | 68 | 61 | 52 | - | 4.78 | 68 | 58 | 51 |
| | ETPNav (An et al., 2024) [TPAMI24] | 11.78 | 3.95 | 72 | 66 | 59 | 11.99 | 4.71 | 65 | 57 | 49 | 12.87 | 5.12 | 63 | 55 | 48 |
| | ETPNav* | 11.35 | 3.93 | 72 | 66 | 59 | 11.40 | 4.69 | 64 | 57 | 49 | 12.72 | 5.10 | 63 | 55 | 48 |
| | **SC²-WM** | 12.15 | 3.85 | 73 | 68 | 60 | 12.05 | 4.60 | 68 | 59 | 51 | 13.88 | 4.97 | 65 | 57 | 50 |
| | HNR (Wang et al., 2024b) [CVPR24] | 11.79 | 3.67 | 76 | 69 | 61 | 12.64 | 4.42 | 67 | 61 | 51 | 13.03 | 4.81 | 67 | 58 | 50 |
| | HNR* | 11.84 | 3.73 | 76 | 69 | 61 | 12.76 | 4.57 | 67 | 61 | 51 | 12.92 | 4.85 | 67 | 58 | 50 |
| | **SC²-WM** | 12.09 | 3.28 | 80 | 71 | 64 | 12.89 | 4.25 | 70 | 66 | 54 | 13.42 | 4.90 | 71 | 62 | 53 |

Note: Following prior work, we report the results with different precision formats across camera configurations——integers for panoramic settings and two decimal places for monocular settings.

**Analysis of Optimization Objectives for Conditional World-Aware Adaptation.** As detailed in Table 6, we investigate the effectiveness of three distinct self-supervised objectives for the adaptation of the world model during inference. Recent test-time adaptation studies on dynamic tasks have demonstrated the effectiveness of different optimization objectives (Guo et al., 2026b; Gao et al., 2025a). We compare the Self-Entropy Loss, which minimizes the entropy of the scoring distribution to encourage high-confidence predictions; the Asynchronous Scoring Loss, which utilizes the posterior score from the subsequent step to supervise the current step under the assumption that later steps possess better context; and our proposed Foresight-Reconstruction Loss (FRL, Eq.(13)), which aligns the World Model's predicted next-step features with the actual observations. Experimental results indicate that the Self-Entropy Loss fails to improve performance, as it lacks grounded supervision and merely reinforces existing beliefs without correcting errors. Besides, the Asynchronous Scoring Loss achieves superior results compared to the w/o-adaptation baseline. However, the improvement is not significant due to the

*Table 5.* Experimental results on RxR-CE datasets. Results better than the base model are shown in blue. Best results for the panoramic and monocular settings are each highlighted in bold.

| Camera | Methods | Val Seen | | | | | | | Val Unseen | | | | | | | Test Unseen | | | | | | |
|---|---|---|---|---|---|---|---|---|---|---|---|---|---|---|---|---|---|---|---|---|---|---|
| | | TL↓ | NE↓ | OSR | SR | SPL | NDTW | SDTW | TL↓ | NE↓ | OSR | SR | SPL | NDTW | SDTW | TL↓ | NE↓ | OSR | SR | SPL | NDTW | SDTW |
| Monocular | LAW (Raychaudhuri et al., 2021) | **7.92** | 11.94 | 20.0 | 7.0 | 6.0 | - | - | **4.01** | 10.87 | 21.0 | 8.0 | 8.0 | - | - | - | - | - | - | - | - | - |
| | CM² (Georgakis et al., 2022) | - | - | - | - | - | - | - | 12.29 | 8.98 | 25.3 | 14.4 | 9.2 | - | - | - | - | - | - | - | - | - |
| | WS-MGMap (Chen et al., 2022a) | 10.37 | 10.19 | 27.7 | 14.0 | 12.3 | - | - | 10.80 | 9.83 | 29.8 | 15.0 | 12.1 | - | - | - | - | - | - | - | - | - |
| | NaVid (Zhang et al., 2024) | - | - | - | - | - | - | - | 10.59 | 8.41 | 34.5 | 23.8 | 32.2 | - | - | - | - | - | - | - | - | - |
| | A²-Nav (Chen et al., 2023) | - | - | - | - | - | - | - | - | - | - | 16.8 | 6.3 | - | - | - | - | - | - | - | - | - |
| | NavMorph (Yao et al., 2025) | 21.61 | 9.80 | 41.27 | 29.81 | 23.23 | 44.51 | 22.68 | 20.28 | 8.85 | 43.05 | 30.76 | 22.84 | 44.19 | 23.30 | 21.13 | 9.81 | - | 24.93 | 16.82 | 33.71 | 15.64 |
| | monoVLN (Lu et al., 2025) | - | - | - | - | - | - | - | - | 8.29 | 37.7 | 31.8 | 26.8 | - | 25.2 | - | - | - | - | - | - | - |
| | VLN-3DFF (Wang et al., 2025c) | - | - | - | - | - | - | - | - | 8.79 | 36.7 | 25.5 | 18.1 | - | - | - | - | - | - | - | - | - |
| | VLN-3DFF* | 18.91 | 9.87 | 40.54 | 27.72 | 20.61 | 42.37 | 20.94 | 16.21 | 9.41 | 38.40 | 26.66 | 20.11 | 42.91 | 20.36 | **20.85** | 10.19 | - | 23.41 | 15.43 | 32.38 | 14.75 |
| | SC²-WM | 22.08 | **9.39** | **44.83** | **31.66** | **23.97** | 42.81 | **23.65** | 20.87 | **8.36** | **48.38** | **35.77** | **27.20** | **44.93** | **26.47** | - | - | - | - | - | - | - |
| Panoramic | Seq2Seq (Anderson et al., 2018b) | - | - | - | - | - | - | - | 7.33 | 12.1 | - | 13.93 | 11.96 | 30.86 | 11.01 | - | 12.10 | - | 13.93 | 11.96 | 30.86 | 11.01 |
| | Reborn (An et al., 2022b) | - | 5.69 | - | 52.43 | 45.46 | 66.27 | 44.47 | - | 5.98 | - | 48.60 | 42.05 | 63.35 | 41.82 | - | 7.10 | - | 45.82 | 38.82 | 55.43 | 38.42 |
| | CWP-CMA (Hong et al., 2022) | - | - | - | - | - | - | - | - | 8.76 | - | 26.59 | 22.16 | 47.05 | - | 20.04 | 10.4 | - | 24.08 | 19.07 | 37.39 | 18.65 |
| | CWP-RecBERT (Hong et al., 2022) | - | - | - | - | - | - | - | - | 8.98 | - | 27.08 | 22.65 | 46.71 | - | 20.09 | 10.4 | - | 24.85 | 19.61 | 37.30 | 19.05 |
| | AO-Planner (Chen et al., 2024) | - | - | - | - | - | - | - | - | 7.06 | - | 43.3 | 30.5 | 50.1 | - | - | - | - | - | - | - | - |
| | LAW-Pano (Raychaudhuri et al., 2021) | 6.27 | 12.07 | 17.0 | 9.0 | 9.0 | - | - | 4.62 | 11.04 | 16.0 | 10.0 | 9.0 | - | - | - | - | - | - | - | - | - |
| | UnitedVLN (Dai et al., 2024) | - | 4.74 | - | 65.1 | 52.9 | 69.4 | 53.6 | - | 5.48 | - | 57.9 | 45.9 | 63.9 | 48.1 | - | - | - | - | - | - | - |
| | NavMorph (Yao et al., 2025) | 20.80 | 5.10 | 67.88 | 64.95 | 54.17 | 70.94 | 54.82 | 21.33 | 5.67 | 66.02 | 58.02 | 48.98 | 64.77 | 48.85 | 23.36 | 6.67 | - | 54.98 | 43.02 | 57.31 | 44.76 |
| | ETPNav (An et al., 2024) | - | 5.03 | - | 61.46 | 50.83 | 66.41 | 51.28 | - | 5.64 | - | 54.79 | 44.89 | 61.90 | 45.33 | - | 6.99 | - | 51.21 | 39.86 | 54.11 | 41.30 |
| | ETPNav* | 18.16 | 5.06 | 64.06 | 62.09 | 50.64 | 66.06 | 51.17 | 18.92 | 5.96 | 63.66 | 54.83 | 44.62 | 61.36 | 44.87 | 21.83 | 6.92 | - | 51.38 | 39.90 | 53.85 | 40.91 |
| | SC²-WM | 18.88 | 4.91 | 67.13 | 64.89 | 52.71 | 68.34 | 52.98 | 19.65 | 5.57 | 65.17 | 56.72 | 46.17 | 63.66 | 46.87 | - | - | - | - | - | - | - |
| | HNR (Wang et al., 2024b) | - | 4.85 | - | 63.72 | 53.17 | 68.81 | 52.78 | - | 5.51 | - | 56.39 | 46.73 | 63.56 | 47.24 | - | 6.81 | - | 53.22 | 41.14 | 55.61 | 42.89 |
| | HNR* | 19.74 | 4.93 | 66.01 | 63.55 | 53.37 | 69.02 | 52.66 | 20.41 | 5.75 | 64.93 | 56.48 | 46.62 | 63.43 | 47.38 | 23.02 | 6.88 | - | 53.33 | 41.18 | 55.47 | 42.95 |
| | SC2-WM | 20.54 | 4.98 | **68.76** | 65.10 | 54.67 | 71.32 | 54.94 | 21.01 | 5.72 | 66.77 | 60.02 | 49.79 | 66.77 | 49.50 | - | - | - | - | - | - | - |

Note: The official evaluation server for the Test Unseen split of RxR-CE dataset is currently unavailable, thus we only report results on the Val Seen and Unseen split.

*Table 6.* Ablation study of different loss functions for online model adaptation, including average time per episode. Best results are highlighted in **bold**.

| Methods | R2R-CE Val Unseen | | | | | | | Time (s) |
|---|---|---|---|---|---|---|---|---|
| | TL↓ | NE↓ | OSR | SR | SPL | NDTW | SDTW | |
| Base model | 26.16 | 6.05 | 54.92 | 43.77 | 29.39 | 40.94 | 29.30 | 17.21 |
| SC²-WM *w/o* Adaptation | 18.73 | 5.46 | 59.22 | 49.86 | 36.05 | 48.10 | 34.62 | 17.78 |
| SC²-WM *w/* Self-Entropy Loss | 18.62 | 5.46 | 58.62 | 49.48 | 35.84 | 48.32 | 34.52 | 19.93 |
| SC²-WM *w/* Async Scoring Loss | 18.57 | 5.42 | **59.43** | 50.19 | 36.20 | 48.31 | 34.78 | 18.76 |
| **SC²-WM *w/* FRL (Ours)** | **17.65** | **5.37** | 58.84 | **50.90** | **37.17** | **49.31** | **35.70** | 18.92 |

scoring function's lack of strict temporal consistency; influenced by extraneous factors like the absolute step index, the posterior score serves as a noisy supervision target. In contrast, FRL achieves superior performance by retaining objective consistency with the training phase. By leveraging the deviation between the model's 'imagination' and reality as a robust error signal, FRL ensures effective model updates and better generalization in unseen environments.

**Ablation Study of Visual Memory Design.** We further investigate the optimal size of the memory buffer, which stores visual features from preceding steps to augment the current observation. We evaluate memory lengths $L \in \{2, 4, 6, 8\}$ and find that $L = 4$ yields the best performance, with results shown in Table 7. Shorter memory lengths (e.g., $L = 2$) provide insufficient historical context, leading to a 'myopic' agent that fails to capture necessary temporal dependencies. Conversely, excessive memory lengths (e.g., $L = 8$) degrade performance due to attention dilution. As the search space expands, the attention mechanism struggles to allocate focus efficiently, resulting in a dispersed (near-uniform) weight distribution that drowns out salient features. Thus, $L = 4$ strikes an optimal balance, providing rich context without overwhelming the attention module.

Additionally, we clarify that VCM is designed as a lightweight functional module tailored for our closed-loop correction framework, rather than a standalone memory architecture. VCM performs query-conditioned retrieval directly in the latent space, avoiding explicit map construction while enabling efficient access to short-term visual history. This design makes it particularly suitable for high-frequency step-wise refinement in real-time VLN settings. The employed 7D geometric encoding follows standard VLN practice, consisting of 4D angular features (sine/cosine of relative heading and elevation) together with 3D distance features.

## D.3. Further Discussions

**Clarification of "Closed-Loop".** Our formulation differs from a classical control-theoretic "closed-loop", which relies on external environmental measurements to compute feedback signals. In our work, "closed-loop" refers to an internal feedback

*Table 7.* Ablation study of different memory lengths in the proposed World Model.

| Methods | R2R-CE Val Unseen | | | | | | |
|---|---|---|---|---|---|---|---|
| | TL $\downarrow$ | NE $\downarrow$ | OSR | SR | SPL | NDTW | SDTW |
| Base model | 26.16 | 6.05 | 54.92 | 43.77 | 29.39 | 40.94 | 29.30 |
| SC$^2$-WM ($L=2$) | 21.47 | 5.96 | 57.31 | 46.49 | 32.24 | 43.29 | 31.17 |
| **SC$^2$-WM ($L=4$)** | 17.65 | **5.37** | **58.84** | **50.90** | **37.17** | **49.31** | **35.70** |
| SC$^2$-WM ($L=6$) | 18.50 | 5.97 | 53.83 | 46.00 | 32.81 | 45.82 | 32.59 |
| SC$^2$-WM ($L=8$) | **16.13** | 5.82 | 51.17 | 44.48 | 33.78 | 48.84 | 32.75 |

cycle over latent states. Specifically, the agent evaluates a provisional action through world-model foresight and computes a discrepancy between the predicted next state and the current state. This signal is then used to refine the latent state before committing to a final decision, forming a self-correcting loop within the model. Importantly, this mechanism operates on internally predicted dynamics rather than external measurements. Therefore, it is more precisely an internal refinement loop grounded in model-based foresight, rather than a classical feedback control process. We will clarify this distinction in final version. Our terminology is inspired by predictive processing in cognitive science (Huang & Rao, 2011), where internal prediction errors are used to guide behavior. This design is particularly motivated by VLN settings (Krantz et al., 2020), where dense and reliable external feedback signals are difficult to obtain, making internal feedback a practical alternative.

**Difference from critic-based feedback.** Our feedback mechanism differs fundamentally from critic-based feedback in both representation and functionality. In reinforcement learning, a critic typically outputs a scalar value estimate that reflects the expected return of the current state or action. Such scalar supervision is highly compressed and, under partial observability, can become noisy or biased. In contrast, the proposed feedback $\Delta_t$ is a high-dimensional latent signal derived from world-model foresight. Rather than merely evaluating whether an action is beneficial, $\Delta_t$ captures structured state-transition information that can directly guide online state correction. Consequently, critic signals mainly serve an evaluative role, whereas $\Delta_t$ functions as a predictive corrective signal. Our goal is therefore not to compare reinforcement learning with world modeling, but to investigate how foresight-driven latent feedback can improve error correction in VLN-CE. Moreover, quantitatively isolating and comparing these heterogeneous signals across paradigms is itself non-trivial.

**Potential extension to reinforcement learning.** The current implementation of SC$^2$-WM adopts imitation learning because VLN-CE exhibits severe partial observability and sparse reward supervision, making purely reward-driven optimization unstable in practice. Demonstration supervision is therefore important for establishing reliable navigation behaviors before introducing exploration. Nevertheless, the proposed framework is naturally compatible with reinforcement learning. For example, the foresight feedback $\Delta_t$ could be incorporated as an intrinsic reward to provide denser training signals, or used as an auxiliary latent representation to facilitate policy optimization. Importantly, the core contribution of SC$^2$-WM—leveraging world-model foresight for online self-correction—is orthogonal to the choice of policy learning paradigm, making reinforcement-learning-based extensions a promising future direction.

# E. Real-World Verification

We further evaluate SC$^2$-WM on a real-world vision-and-language navigation platform using a Unitree GO2 quadruped robot equipped with an Intel RealSense D435i camera, as illustrated in Figure 5. Experiments are conducted in indoor home environments, covering an area of approximately $100m^2$ and includeing diverse furniture layouts/rooms. The robot is required to follow natural language navigation instructions such as "*Go through the table and find the trash can*". The robot executes actions (forward, turn left/right, stop) consistent with our simulation setup, and visual observations are processed in real-time. Depth sensing is utilized only for collision avoidance as a safety mechanism.

We perform 100 navigation trials (500 physical runs, 100 trials $\times$ 5 repetitions) across diverse layouts and instructions with varying levels of complexity. A trial is considered successful if the robot reaches the goal location described in the instruction (within 1.5 meters). SC$^2$-WM achieves an overall success rate of 85%, compared to 70% for the baseline. The performance gain is particularly notable in scenarios involving visual ambiguity or decision-making at intersections, where the self-correction mechanism enables the robot to recover from suboptimal actions. Importantly, the proposed adaptation mechanism introduces only limited runtime overhead in practice. Since adaptation is conditionally triggered rather than

applied at every step, its frequency remains low. In addition, we adopt an asynchronous execution strategy, where gradient updates in Eq. (14) are performed in parallel while the robot executes the current action, preventing the control loop from waiting for backpropagation. Although our current system relies on offboard computation, the lightweight design remains compatible with resource-constrained embodied platforms. Overall, these results demonstrate that SC$^2$-WM transfers effectively from simulation to real-world deployment and exhibits promising practicality for real-world VLN tasks.

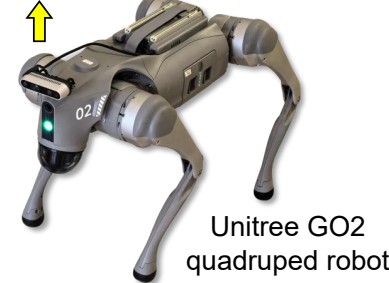

**Instruction:** *"Go through the table and find the trash can. Turn right and stand besides the TV."*

Intel RealSense D435i Camera

Unitree GO2 quadruped robot

*Figure 5.* Real-world deployment of SC$^2$-WM on a quadruped robot navigating indoor environments with natural language instructions.

