# OpenReview forum: "SC$^{2}$-WM: A Self-Correcting World Model with Closed-Loop Feedback for Vision-and-Language Navigation in Continuous Environments"
_ICML.cc/2026/Conference — ICML 2026 regular_

### Official Review · Reviewer_Byth · 2026-03-11

**Soundness:** 3
**Presentation:** 3
**Significance:** 3
**Originality:** 4
**Overall Recommendation:** 5
**Confidence:** 4

**Summary:**

The paper proposes SC2-WM, a self-correcting world model framework for Vision-and-Language Navigation in Continuous Environments (VLN-CE). To address the error accumulation issue in conventional open-loop execution, the authors introduce an internal feedback mechanism derived from the world model's foresight predictions. The framework features a dual-level self-correction mechanism: (1) Feedback-guided plan refinement for state-level correction prior to action execution, and (2) Conditional world-aware adaptation for model-level parameter updates at test time. The method demonstrates strong empirical performance across R2R-CE and RxR-CE benchmarks and includes a real-world deployment on a quadruped robot.

**Compliance With Llm Reviewing Policy:**

Affirmed.

**Final Justification:**

I would like to thank the authors for their detailed and thoughtful rebuttal. The clarifications regarding the term "closed-loop", the justification for the local linearity approximation in the latent space, and the detailed explanation of the asynchronous execution strategy for real-world deployment are all very clear and convincing. These explanations have fully addressed my initial concerns. I will give a higher score, as I believe this work is a solid and valuable contribution to the VLN-CE community.

**Key Questions For Authors:**

1. Conceptual Clarity on "Closed-Loop" and "State Correction": The paper describes the framework as a "closed-loop" system performing "state-level correction". However, in classical control theory, a closed-loop system fundamentally relies on actual environmental measurements to compute error signals. In your framework, the feedback signal Δt is derived entirely from the internal world model's foresight rather than real environmental observations. Could you clarify this distinction? Would it be more accurate to describe this as a model-based internal refinement loop (or mental rehearsal) rather than a true closed-loop control process?
	2. Inference Latency of Online Updates in Real-World Deployment: The real-world deployment on the Unitree GO2 platform (Section 4.3 and Appendix E) is an excellent addition. However, the conditional world-aware adaptation requires online gradient updates via backpropagation (Eq. 14). While Table 6 reports episode times measured on an RTX 3090 GPU, edge computing hardware typically used on quadrupeds (e.g., Jetson series) has significantly less compute. How much single-step inference delay does this online parameter update introduce during real-world deployment? Does the robot need to pause frequently to wait for these gradient updates, and how does this impact the real-time requirements of continuous control?

**Limitations:**

yes

**Strengths And Weaknesses:**

strengths:
	Originality (Elegant Theoretical Motivation): The approach of using predictive foresight to generate internal feedback is highly innovative. Inspired by predictive processing in cognitive science, it offers a clever workaround for the sparsity of external reward signals in embodied navigation tasks.
	Soundness (Comprehensive Empirical Validation): The experimental design is rigorous. The method is evaluated on both R2R-CE and RxR-CE datasets under both panoramic and monocular settings. The ablation studies meticulously disentangle the contributions of the State-C, VCM, and Model-C components.
	Soundness (Sim-to-Real Deployment): Deploying the framework on a physical Unitree GO2 quadruped robot and achieving an 85% success rate in real-world indoor environments is a highly commendable effort that significantly bridges the sim-to-real gap.
	Presentation: The paper is exceptionally well-written. Figure 1 provides a crystal-clear visual contrast between conventional open-loop VLN, classical closed-loop control, and the proposed internal feedback mechanism.
weaknesses:
	Presentation (Conceptual Overclaim on "Closed-Loop"): The paper heavily markets the framework as a "closed-loop" system. However, in classical control theory, closed-loop systems require actual feedback from the external environment. The proposed feedback signal is derived entirely from the internal world model's foresight, making it technically an internal mental rehearsal or latent plan refinement, rather than a true closed-loop state correction.
	Soundness (Mathematical Rigor in Latent Space): The internal feedback relies on a simple linear subtraction in the latent space (~s_t+1 - st). The mathematical justification for performing Euclidean linear operations within a presumably highly non-linear latent manifold is somewhat lacking.
	Soundness (Inference Latency in Real-world Deployment): While the conditional world-aware adaptation (Model-C) brings performance gains, the necessity of performing online gradient updates (Eq. 14) via backpropagation poses concerns regarding inference latency, especially on edge computing devices typical of robotic platforms.

---

> ### Author Rebuttal · Authors · 2026-03-31
>
> 1. **Clarification of "Closed-Loop".**
> We thank the reviewer for clarification. We acknowledge that our formulation differs from a classical control-theoretic "closed-loop", which relies on external environmental measurements to compute feedback signals.
> In our work, "closed-loop" refers to an internal feedback cycle over latent states. Specifically, the agent evaluates a provisional action through world-model foresight and computes a discrepancy $Δ_t$ between the predicted next state and the current state. This signal is then used to refine the latent state before committing to a final decision, forming a self-correcting loop within the model.
> Importantly, this mechanism operates on internally predicted dynamics rather than external measurements. Therefore, it is more precisely an internal refinement loop grounded in model-based foresight, rather than a classical feedback control process. We will clarify this distinction in final version.
> Our terminology is inspired by predictive processing in cognitive science (Lines 60–76), where internal prediction errors are used to guide behavior. This design is particularly motivated by VLN settings (Lines 48–50), where dense and reliable external feedback signals are difficult to obtain, making internal feedback a practical alternative.
> &nbsp;
>
> 2. **Rigor in latent space.**
> We thank the reviewer for this important point.
> Our design is based on a common approximation in representation learning: learned latent spaces are locally smooth and can be reasonably linearized within a small neighborhood. In our case, $Δ_t = s̃_{t+1} − s_t$ captures a local discrepancy between the current state and its one-step foresight prediction, rather than a global geometric relation.
> Importantly, $Δ_t$ serves as an operational signal for state refinement, rather than as a precise metric of distance on the latent manifold. This local difference has proven sufficient to guide effective correction in practice. This is consistent with established practices in latent dynamics modeling [1,2], where learned latent spaces are sufficiently smooth such that simple vector differences can effectively capture short-term temporal transitions. Similar assumptions of local linearity have also been explored in prior work on latent dynamics [3].
> We adopt this simple formulation for efficiency and stability, and exploring more structured alternatives (e.g., geometry-aware or manifold-based operations) is an interesting direction for future work.
>
> &nbsp;&nbsp; &nbsp;&nbsp; *Reference:*
> &nbsp;*[1] Hafner, D., et al. "Learning latent dynamics for planning from pixels." ICML (2019).*
>
> &nbsp;&nbsp;&nbsp;&nbsp;&nbsp;&nbsp; *[2] Hafner, D., et al. "Dream to control: Learning behaviors by latent imagination." ICLR (2020).*
>
> &nbsp;&nbsp;&nbsp;&nbsp;&nbsp;&nbsp; *[3] Watter, et al. "Embed to control: A locally linear latent dynamics model for control from raw images." Neurips (2015).*
> &nbsp;
>
> 3. **Inference latency of online adaptation**
> We thank the reviewer for raising this important practical concern. In our real-world deployment, the online adaptation does not block control execution.
>
> - First, the adaptation is **conditionally triggered** rather than applied at every step, which significantly reduces its frequency. More importantly, we adopt an **asynchronous execution strategy**, decoupling the control loop from the adaptation process: gradient updates (Eq. 14) are performed in parallel while the robot executes the current action. As a result, the control loop never pauses to wait for backpropagation.
>
> - Empirically, in our real-world deployment, the robot communicates via a portable Wi-Fi module with a remote server equipped with an RTX 4090 GPU. Under this setup, model inference is lightweight, taking ~0.08s per step for SC²-WM compared to ~0.06s for the baseline. In contrast, the overall step latency is dominated by sensing, communication, and physical execution: due to safe motion constraints (0.2 m/s linear, 0.2 rad/s angular), a single navigation step takes ~7.4s for SC²-WM and ~7.2s for the baseline.
> This indicates that the additional computation introduced by SC²-WM has minimal impact on the control-critical path, as most of the latency is governed by physical execution rather than model inference.
>
> - Regarding edge deployment, while our current experiments use offboard computation, the conditional and asynchronous design keeps the additional overhead limited, making it compatible with resource-constrained settings. We will include further discussion in the appendix.

---

> > ### Author Rebuttal · Reviewer_Byth · 2026-04-03
> >
> > I would like to thank the authors for their detailed and thoughtful rebuttal. The clarifications regarding the term "closed-loop", the justification for the local linearity approximation in the latent space, and the detailed explanation of the asynchronous execution strategy for real-world deployment are all very clear and convincing. These explanations have fully addressed my initial concerns. I will give a higher score, as I believe this work is a solid and valuable contribution to the VLN-CE community.

---

> > > ### Author Response · Authors · 2026-04-07
> > >
> > > We sincerely thank the reviewer for the positive feedback.
> > >
> > > We are pleased that our clarifications helped address the concerns. We will carefully incorporate all the discussed revisions into the final version to further strengthen the clarity and overall presentation of the paper.
> > >
> > > We greatly appreciate the reviewer’s time and constructive feedback.

---

### Official Review · Reviewer_ShMC · 2026-03-12

**Soundness:** 3
**Presentation:** 3
**Significance:** 2
**Originality:** 2
**Overall Recommendation:** 4
**Confidence:** 4

**Summary:**

This paper proposes SC2-WM, a self-correcting world model framework for Vision-and-Language Navigation in Continuous Environments (VLN-CE). The core idea is to derive internal feedback signals from world-model foresight predictions to enable closed-loop decision-making without relying on sparse external rewards. The method introduces a dual-level correction mechanism: (1) feedback-guided plan refinement that modulates latent states before action execution, and (2) conditional world-aware adaptation that selectively updates world model parameters at test time when feedback indicates model capacity insufficiency. Experiments on R2R-CE and RxR-CE benchmarks demonstrate improvements over several baselines, with additional real-world deployment on a quadruped robot.

**Compliance With Llm Reviewing Policy:**

Affirmed.

**Final Justification:**

The rebuttal addressed my main concerns sufficiently, especially by clarifying the feedback mechanism, implementation details, and practical overhead, which leads me to update my score from 3 to 4. Overall, I find the paper sound and reasonably original, with a meaningful contribution to self-correcting navigation under partial observability. I still have some reservations about the practical robustness of the memory component, since performance is best at L=4 and degrades for larger values, but I no longer view this as a decisive weakness.

**Key Questions For Authors:**

- Theoretical comparison to critic-based feedback: Could you provide analysis or empirical comparison showing why world-model foresight feedback (Δt) offers advantages over a learned critic's value estimates for detecting/correcting decision errors? How do the two signals differ in terms of temporal scope, bias, or robustness to partial observability? A convincing answer could strengthen the soundness assessment.
- Disentangling agent vs. environment dynamics in Δt: Equation 9 computes latent discrepancy assuming action-induced changes. In partially observable settings, how does the method handle cases where observation changes stem from exogenous factors rather than the agent's action? Could this introduce misleading feedback signals? Clarification here would address a key soundness concern.
- Extension to pure RL settings: The training loss (Eq. 11) uses expert actions. Could SC²-WM operate in settings without demonstrations, relying solely on sparse task rewards? If so, how would the feedback signal be integrated with RL objectives? This addresses the method's generality beyond imitation learning.

**Limitations:**

- Computational overhead: The dual policies (π, π̃), VCM memory, and conditional adaptation likely increase inference latency. Quantifying this overhead and discussing trade-offs for real-time deployment would be valuable.
- Memory length sensitivity: Table 7 shows performance degrades when L > 4, suggesting the method is sensitive to hyperparameter choice. A discussion of how to select L adaptively or robustly would strengthen practical guidance.

**Strengths And Weaknesses:**

Strengths:

- Comparison to large number of baselines and showing good performance
- Ablation studies systematically isolate contributions of VCM, state-level correction, and model-level adaptation.
- Addresses a genuinely important problem: mitigating error accumulation in partially observable tasks
- The dual-level correction concept (state + model) could inspire similar designs in other embodied AI tasks where external rewards are sparse.

Weaknesses:

- Formula 9 concern: Δt = s̃t+1 − st assumes state changes are solely action-induced, but in partially observable environments, exogenous factors (e.g., new objects entering view) can alter observations independently. The paper does not address how the feedback signal disentangles agent-caused vs. environment-caused changes.
- ψ network opacity: Equation 10 describes ψ as a "learnable neural network" but provides no architectural details, initialization strategy, or training dynamics. This limits reproducibility and understanding of how state refinement actually occurs.
- Table 6 results: Adaptation gains (e.g., SR: 49.86% → 50.90%) appear marginal and potentially within experimental variance. Statistical significance testing or confidence intervals would help assess robustness.
- Figure 2 complexity: The framework diagram is dense with many interacting components (VCM, π, π̃, fp, fq, ψ, adaptation triggers). A simplified flowchart or pseudocode would significantly improve accessibility.
- Trigger mechanism clarity: The conditional adaptation criteria (κt exceeding τ-th percentile, entropy < ϵ) are described heuristically. More intuitive explanation or visualization of when/why adaptation fires would help.

---

> ### Author Rebuttal · Authors · 2026-03-31
>
> 1. **Exogenous Factors.** Thanks for insightful point. We do not explicitly disentangle action-induced and exogenous changes. Instead, the observation $o_t$, obtained after executing $a_{t-1}$, inherently reflects both agent-driven and environment-driven variations. Our world model learns state transitions via a reconstruction loss (Eq.13) supervised by $o_t$. Thus, the latent state encodes both sources of change, and $\Delta_t$ represents a discrepancy in the *overall predicted transition*, rather than purely action-induced effects. Disentanglement is a promising future direction, while our focus is on leveraging such internal signals for feedback-based correction.
> 2. **Details of ψ network.** ψ (Eq.10) is implemented as a dynamic gating network that predicts a residual correction conditioned on reconstructed state and action features. A sigmoid gate modulates the update magnitude adaptively. The network is trained end-to-end with the same objectives; details will be added in the appendix.
> 3. **Marginal Gains in Tab.6.** Results are averaged over 5 runs due to stochasticity introduced by adaptation. Full per-run results ([Tab.R1](https://anonymous.4open.science/r/ICML2026-Rebuttal-6618-EA9F/README.md)) is provided to show consistent improvements beyond typical variance.
> 4. **Clarity of Fig.2.** Thanks for valuable feedback. To complement the pipeline, we provide a concise pseudocode of SC²-WM in [Fig. R2](https://anonymous.4open.science/r/ICML2026-Rebuttal-6618-EA9F/README.md#figure-r2) and will include in the final version.
> 5. **Details of Trigger Mechanism.** We use a dynamic percentile-based queue over recent $\kappa_t$, triggering adaptation when it exceeds the τ-th percentile (τ = 0.5). This self-adapts to recent dynamics without manual threshold tuning. We will clarify this in the appendix.
> 6. **Comparison to critic-based feedback.** We appreciate this insightful suggestion. The key difference lies in the form and role of the signals. A critic provides a scalar value at each step, indicating merely the absolute magnitude or intensity of expected returns. Under partial observability, such compressed, absolute estimates can be noisy or biased. In contrast, our feedback
> $Δ_t$ is a high-dimensional latent vector derived from foresight. Thus, the two signals serve different roles: a scalar critic is evaluative (indicating whether an error occurred), whereas $Δ_t$ preserves structured state information, making it more informative and actionable for online correction. We respectfully clarify that our focus is on understanding world-model-based feedback for error correction, rather than making a comparison between RL and world models. Moreover, directly isolating and quantitatively evaluating the quality of these two signals across paradigms is non-trivial.
> 7. **Extension to RL.** We appreciate this insightful question.  SC²-WM relies on demonstrations (IL) because extreme partial observability and sparse rewards in VLN-CE make purely reward-driven optimization unstable. IL is essential to establish meaningful navigation behaviors first. However, our method can be extended to RL. In such settings, the feedback $Δ_t$ could serve as an intrinsic reward to provide dense signals, or as an auxiliary representation to accelerate policy optimization. But our core contribution—world-model foresight for error correction—is fundamentally orthogonal to RL, which remains a promising future direction.
> 8. **Computational overhead.** In real-world setup (robot via portable Wi-Fi to a remote RTX 4090 server), model inference takes \~0.08 s/step (baseline: \~0.06 s/step). Because the robot operates under strict safety constraints (0.2 m/s linear, 0.2 rad/s angular), total step latency is dominated by sensing, communication, and physical motion (\~7.4s vs. \~7.2s), which remains modest and practically acceptable given the performance gains. Furthermore, we adopt an asynchronous deployment strategy—performing model-level adaptation concurrently with physical execution—to effectively hide inference latency and ensure continuous navigation. We will add this analysis.
> 9. **Sensitivity of $L$.** We agree that varying $L$ causes performance fluctuations ([Tab. R2](https://anonymous.4open.science/r/ICML2026-Rebuttal-6618-EA9F/README.md)). Small $L$ limits useful context, while excessively large $L$ introduces noise; moderate values (e.g., $L=4$) yield the best balance. As $L$ increases, the model leverages more extended historical context, making it more conservative and prone to stop earlier at historically familiar states. Consequently, Trajectory Length (TL) decreases. This reduced exploration naturally lowers the Oracle Success Rate (OSR), as the agent wanders less and has fewer accidental goal passes. Importantly, despite this sensitivity, SC²-WM consistently outperforms the baseline across most metrics for $L \in [2, 8]$. This observation inspires future work to adaptively adjust memory size based on environmental changes.

---

> > ### Author Rebuttal · Reviewer_ShMC · 2026-04-03
> >
> > I have read the author rebuttal. The authors addressed my main concerns sufficiently for me to update my score from 3 to 4. I still have some reservations about the practical usefulness of the memory component, since the best result is achieved at L=4 and increasing L degrades some metrics, but I no longer consider this a decisive weakness.

---

> > > ### Author Response · Authors · 2026-04-07
> > >
> > > We sincerely thank the reviewer for the positive feedback.
> > >
> > > We are pleased that our rebuttal addressed the main concerns. We also appreciate the reviewer’s insightful comments on the memory component, and we will further clarify its design and behavior in the final version to improve clarity and completeness.
> > >
> > > We greatly appreciate the reviewer’s time and valuable feedback.

---

### Official Review · Reviewer_mWT9 · 2026-03-12

**Soundness:** 2
**Presentation:** 3
**Significance:** 3
**Originality:** 3
**Overall Recommendation:** 4
**Confidence:** 4

**Summary:**

The paper proposes a self-correcting world model (SC²-WM) framework for Vision-and-Language Navigation in Continuous Environments (VLN-CE), introducing internal feedback for closed-loop decision making to address state drift and mitigate error accumulation in existing open-loop VLN paradigms. Specifically, the method employs state-level plan refinement prior to action execution and model-level test-time adaptation to handle unseen environments. Experiments on the R2R-CE and RxR-CE benchmarks show consistent improvements in success rate (SR) and path efficiency (SPL) over recent baselines, alongside read-world deployment on a  Unitree GO2 quadruped robot.

**Compliance With Llm Reviewing Policy:**

Affirmed.

**Final Justification:**

Thank the authors for their detailed and well-structured rebuttal. The additional experiments and clarifications provided have adequately addressed my major concerns. I am satisfied with the responses and look forward to seeing the promised revisions incorporated into the final version of the paper.

**Key Questions For Authors:**

1. Can the authors elaborate on how the "Conditional World-Aware Adaptation" actually benefit real-world deployment? Given the significant sim-to-real distribution shift, providing concrete real-world examples or a brief physical ablation of this module would make the claims of its practical utility much more solid.
2. (Related to Weakness 2) Given that inference speed is critical for real-time navigation in VLN-CE, could the authors report the exact inference latency (e.g., FPS or latency per step) of SC²-WM and provide a direct comparison with the baselines?
Other questions please see weaknesses.

**Limitations:**

yes

**Strengths And Weaknesses:**

Strengths：
1. The idea of tackling  state drift and mitigating error accumulation in open-loop VLN by extracting internal feedback signals  via a world model is novel and promising.
2. The proposed methodology is technically sound, with well-motivated and logically designed internal components and correction modules.
3. The empirical results are strong. The approach consistently outperforms recent baselines across multiple benchmarks, and the ablation studies effectively validate the contribution of each individual module.

Weaknesses:
1. The paper's central claim — that SC²-WM mitigates "error accumulation" and "state drift" — lacks direct quantitative support. While the qualitative visualizations in Figure 3 kindly illustrate improved trajectory alignment for two specific episodes, these, alongside aggregate metrics (e.g., SR, SPL, SDTW), primarily indicate general performance gains rather than robustly proving a reduction in temporal error compounding. To convincingly support this claim, the authors should provide either targeted quantitative evidence (e.g., showing the performance gap widening as trajectory length or execution steps increase) or a more comprehensive qualitative/statistical analysis of step-wise deviation across varying instruction complexities. Without such dedicated analysis, the claim of mitigated drift remains empirically insufficiently supported.
2. The paper emphasizes the method's effectiveness in continuous environments, which inherently demand real-time or near-real-time control. While the authors provide a qualitative real-world demonstration, the manuscript lacks any quantitative analysis regarding inference latency, frames per second (FPS), or the overall computational overhead introduced by the world model. Even with a physical deployment, reporting these critical metrics is essential. Without them, it is impossible to rigorously evaluate the system's true control frequency and computational efficiency. This omission significantly undermines the completeness and soundness of its application to continuous navigation.
3. While I acknowledge that the authors treat "Conditional World-Aware Adaptation" (Model-C) as an algorithmic component and have isolated its specific contribution in the ablation study (Table 3), I still have concerns regarding the fairness of the main SOTA comparisons in Table 1. In the main evaluations, the full SC²-WM leverages online test-time adaptation (TTA), whereas the compared baselines are evaluated under the standard frozen-weight paradigm. Pitching a model that actively updates parameters during testing against frozen baselines creates a fundamentally unequal setting. To ensure a completely fair head-to-head architectural comparison, the authors should explicitly include the performance of SC²-WM without TTA (i.e., the variant without Model-C) in the main table, or alternatively, compare the full model against other baselines equipped with a similar online updating mechanism.

---

> ### Author Rebuttal · Authors · 2026-03-31
>
> 1. **Quantitative Analysis on Mitigating Error Accumulation.** We split the validation set into 3 buckets (Short, Medium, Long) by GT trajectory length. As shown in the table below, as trajectories extend, baseline SR drops from 50.2 to 35.6 due to compounded state drift. SC²-WM effectively alleviates this issue: it achieves the highest absolute (+7.2 SR / +9.2 SPL) and relative (+20.2% SR / +35.8% SPL) gains on the longest trajectories (>10.0 m). These results provide compelling evidence that our closed-loop mechanism effectively curtails temporal error accumulation. We will add this detailed analysis to the final version.
> | Trajectory Length (m) | Baseline (SR / SPL) | Ours (SR / SPL) | $Δ$ (SR / SPL) | Relative $Δ$ (SR / SPL) |
> | :--- | :---: | :---: | :---: | :---: |
> | Short (<7.7, Avg. 6.4) | 50.2 / 31.5 | 55.9 / 37.9 | +5.7 / +6.4 | +11.4% / +20.3% |
> | Medium (7.7-10.0, Avg. 8.8) | 47.0 / 30.7 | 54.0 / 39.7 | +7.0 / +9.0 | +14.9% / +29.3% |
> | Long (>10.0, Avg. 12.3) | 35.6 / 25.7 | 42.8 / 34.9 | **+7.2 / +9.2** | **+20.2% / +35.8%** |
>
> Regarding instruction complexity, quantifying it directly is difficult due to linguistic variability. However, the RxR-CE dataset involves longer, multilingual instructions (Lines 301–305), where our method also shows clear improvements, further validating its effectiveness under more complex and extended navigation scenarios.
>
> 2. **Latency and Efficiency.** We thank the reviewer for highlighting this important aspect, which is critical for evaluating real-time feasibility. In *simulation*, we evaluated the computational efficiency on a single NVIDIA RTX 3090 GPU (Tab.6 and the table below). While SC²-WM introduces an additional world model and correction mechanism, the per-step inference remains efficient at **0.082s**, compared to the baseline's 0.066s.
> | Model | Avg. Time / Sample (s) | Avg. Time / Step (s) |
> | :--- | :---: | :---: |
> | Baseline | 17.21 | 0.066  |
> | Ours | 18.17 | 0.082 |
>
> In *real-world deployment*, the robot communicates via Wi-Fi with a remote RTX 4090 server. The overall step latency, however, is dominated by sensing, communication, and physical execution. Due to safe motion constraints (0.2 m/s linear, 0.2 rad/s angular), a single navigation step takes ~7.4s for SC²-WM and ~7.2s for the baseline—a marginal increase that is highly acceptable given the significant performance gains. Crucially, we adopt an **asynchronous execution strategy**: online test-time adaptation runs in parallel with physical movement, ensuring the control loop never stalls for backpropagation. Details will be added to the appendix.
>
> 3. **Fairness of SOTA Comparison.** We thank the reviewer for the careful consideration. We would like to clarify that our online adaptation operates strictly on unlabeled observation streams within the current episode. It does not require any additional annotations, environment pre-exploration, or privileged information. This paradigm of test-time adaptation using purely online streams is well-established and widely accepted in embodied navigation, as seen in prior works such as FSTTA [1] and ATENA [2].
> Furthermore, to provide a strictly fair frozen-to-frozen comparison against existing SOTA methods, we evaluated SC²-WM with the online adaptation module disabled. As shown in Tab. 6, even in this frozen state, our model achieves 59.22 OSR / 49.86 SR / 36.05 SPL, outperforming the frozen baseline (54.90 / 43.80 / 29.40) by a significant margin (+6.06 SR / +6.65 SPL). This demonstrates that our performance gains are not solely reliant on test-time adaptation; rather, the proposed world model architecture inherently learns superior, more robust representations during the training phase. We will explicitly include these comparison results in Tab. 1 of the final version.
>
> &nbsp;&nbsp;&nbsp;&nbsp; *Reference: [1] Gao., et al, "Fast-Slow Test-time Adaptation for Online Vision-and-Language Navigation." ICML (2024).*
>
> &nbsp;&nbsp;&nbsp;&nbsp;&nbsp;&nbsp; *[2] Ko, Heeju, et al. "Active Test-time Vision-Language Navigation." Neurips (2025).*
>
> 4. **Practical Benefit of Conditional World-Aware Adaptation.** This module is fundamentally designed to mitigate general distribution shifts encountered when the agent navigates novel, unseen environments. Thus, it proves exceptionally well-suited for tackling the sim-to-real gap during physical deployment—such as varying lighting conditions and sensor noise. We conducted a real-world ablation study: disabling the online adaptation resulted in an ~4% absolute drop in Success Rate. This demonstrates that continuously aligning the agent's internal world model with the actual physical environment is crucial for maintaining robust real-world performance.
> Furthermore, **this adaptation runs asynchronously with respect to the control loop**: conditional world-aware adaptation is performed in parallel while the robot executes the current action.  We will include additional analysis and qualitative examples in the appendix.

---

> > ### Author Rebuttal · Reviewer_mWT9 · 2026-04-02
> >
> > Thank the authors for their detailed and well-structured rebuttal. The additional experiments and clarifications provided have adequately addressed my major concerns. I am satisfied with the responses and look forward to seeing the promised revisions incorporated into the final version of the paper.

---

> > > ### Author Response · Authors · 2026-04-07
> > >
> > > We sincerely thank the reviewer for the positive feedback and for acknowledging that the rebuttal has addressed the major concerns.
> > >
> > > We are glad that the additional experiments and clarifications were helpful. We will carefully incorporate all the promised revisions into the final version to further improve the clarity, completeness, and overall quality of the paper.
> > >
> > > We also appreciate the reviewer’s support and consideration.

---

### Official Review · Reviewer_wcqu · 2026-03-13

**Soundness:** 2
**Presentation:** 2
**Significance:** 2
**Originality:** 2
**Overall Recommendation:** 4
**Confidence:** 4

**Summary:**

This paper proposes SC2-WM, a self-correcting world model framework for Vision-and-Language Navigation in Continuous Environments (VLN-CE). The authors attempt to focus on an important concept: transforming open-loop VLN agents into closed-loop systems via internally generated feedback signals derived from world-model foresight. The framework introduces two complementary mechanisms: (1) Feedback-Guided Plan Refinement — a state-level correction that modulates the current latent state using discrepancy signals between the current state and a foresighted future state prior to action execution; and (2) Conditional World-Aware Adaptation — a model-level test-time adaptation that selectively updates world model parameters when feedback signals indicate high reliance on foresight guidance (triggered by KL divergence between pre- and post-refinement action distributions). A Visual Calibration Module (VCM) with episodic memory is introduced to stabilize latent representations. Experiments are conducted on R2R-CE and RxR-CE benchmarks under both panoramic and monocular settings, supplemented by real-world deployment on a Unitree GO2 quadruped robot.

**Compliance With Llm Reviewing Policy:**

Affirmed.

**Final Justification:**

Thank the authors for their detailed response. My question has been solved.

**Key Questions For Authors:**

N/A

**Limitations:**

yes

**Strengths And Weaknesses:**

Strengths:

1.	The paper clearly identifies a genuine limitation in existing VLN-CE methods： their open-loop nature, and draws a principled analogy to classical closed-loop control theory and predictive coding in cognitive science. The motivation is coherent and the identified gap is real.

2.	The decomposition of self-correction into state-level (latent state refinement) and model-level (test-time adaptation) is a well-structured design. The two levels are complementary and address different failure modes (immediate inference drift vs. long-term domain shift), which is a thoughtful system design.

3.	The entire framework trains on a single NVIDIA RTX 3090 in under 40 hours.  A compelling argument for practical adoption compared to MLLM-based approaches. This positions the work well as a lightweight alternative.

Weaknesses:

1.	The "World Model" Contribution is Overstated / Poorly Differentiated. The world model here is essentially a recurrent latent-space model with a Transformer-based prior transition and a visual reconstruction objective. The paper positions itself as a novel world-model architecture, but the core novelty lies in how the model is used (closed-loop feedback), not in the model itself. The architectural description of f_p and f_q lacks sufficient technical depth (e.g., hidden dimensions, number of layers, stochastic vs. deterministic components are not fully specified in the main text).

2.	Panoramic Setting Gains are Marginal and Inconsistent. While monocular gains are impressive, panoramic improvements over strong baselines are modest: On R2R-CE Test Unseen (panoramic, HNR base): SR 58→62%, SPL 50→53% — small margins that may be within noise. On RxR-CE Val Unseen (panoramic, HNR base): SR 56→60%, which is stronger, but the Test Unseen server is reported as "currently unavailable," limiting conclusiveness. Compared to NavMorph (panoramic), SC2-WM does not consistently outperform across all metrics (e.g., NavMorph achieves 70 SR on R2R-CE Val Seen vs. SC2-WM's 71, but NavMorph achieves higher SPL in some settings).

3.	VCM Design Choices are Insufficiently Justified. The episodic memory uses cross-attention with temporal decay (Eq. 8). However: The memory size L=4 is chosen by grid search, but the performance drop from L=4 to L=6 and L=8 is dramatic (SR: 50.9 → 46.0 → 44.5), which is surprisingly non-monotonic. The paper attributes this to "attention dilution" but provides no analysis of the attention weight distributions. The spatial-temporal encoding (Appendix C) uses a 7-dimensional geometric feature. Why 7 dimensions specifically? The choice seems arbitrary. VCM is clearly inspired by NetVLAD (Arandjelovic et al., 2016) and transformer-based memory, but the novelty over prior work on episodic memory in VLN (e.g., GridMM, BEVBert) is not clearly articulated.

4.	Real-World Experiments Lack Rigor. 100 trials are conducted but the breakdown by instruction complexity, environment layout, and failure mode is absent. No statistical significance (e.g., confidence intervals) is reported for the 85% vs. 70% comparison. The instructions shown in Figure 4 are quite simple (2–3 landmarks). Harder instructions with backtracking or multiple turns — the scenarios the system claims to handle — are not demonstrated. No ablation in the real world to confirm that the correction mechanism (vs. baseline) is responsible for the improvement.

5.	Qualitative Analysis is Limited. Figure 3 shows only two navigation episodes. There is no failure case analysis. When does SC2-WM fail to self-correct? Understanding the limitations of the feedback mechanism is as important as demonstrating successes.

---

> ### Author Rebuttal · Authors · 2026-03-31
>
> 1. **World Model & Contribution.** We follow a standard latent dynamics formulation (e.g., RSSM) and do not claim architectural novelty. Our contribution is a closed-loop self-correction mechanism (Eq.1–7) that enables the agent to simulate future states, compute latent discrepancies from provisional actions, and refine decisions before execution, which transforms conventional open-loop prediction into feedback-driven decision refinemen. Importantly, **this feedback mechanism is not a superficial modification but fundamentally changes how the world model is utilized—from passive prediction to active decision refinement—leading to improved robustness in VLN-CE**.Both $f_p$ and $f_q$ are implemented as 4-layer Transformer-based latent transition modules (hidden size 768) with stochastic variables trained via a KL objective (Eq.12). We will add these details.
> 2. **Panoramic Performance & Comparisons.** Our method consistently improves over the HNR baseline across both monocular and panoramic settings. On R2R-CE Test Unseen, we achieve +4% SR and +3% SPL, which are comparable to or larger than recent methods (NavMorph [ICCV 25]: +2% SR/SPL; g3d-lf [CVPR 25]: +0% SR, +1% SPL), indicating that they are meaningful rather than noise. Larger gains in monocular environments are expected due to stronger partial observability, where feedback-based correction plays a more critical role *(also observed in NavMorph)*, but remain consistent in panoramic cases, indicating robustness beyond extreme settings, demonstrating its effectiveness beyond extreme partial observability. The RxR-CE Test server is unavailable, so we report Val Unseen following recent works (MonoDream [AAAI'26], HSAN [NeurIPS'25]). We also clarify that our reported NavMorph results are HNR-based. Under a fair comparison on R2R-CE Val Seen, SC²-WM (64% SPL) outperforms NavMorph (62%). We will clarify this in the revision.
> 3. **VCM Design.** We analyze attention distributions across memory sizes. As $L$ increases from 4 (default) to 8, average attention on the critical recent step ($t-1$) drops from 55% to 22%, while entropy rises (1.13 to 1.99). This dilution effect weakens the model's ability to prioritize recent observations critical for correction, explaining the performance drop at larger $L$.
> The 7D encoding is aligned with standard VLN geometric features, consisting of 4D angular (sin/cos of relative heading and elevation) and 3D distance features. We clarify that VCM is a functional module tailored for our closed-loop system. The choice over prior episodic memories (GridMM, BEVBert) is driven by the need for efficient, step-wise correction. While prior works build explicit spatial maps (grid/BEV) that introduce overhead and pose sensitivity, VCM performs query-conditioned retrieval directly in the latent space. This avoids explicit map construction while enabling efficient access to short-term history. Thus, its novelty lies in adapting memory for closed-loop local refinement, rather than proposing a new architecture.
> 4. **Real-World Evaluation.** The use of 2–3 landmarks reflects the current practical scope of real-world VLN under physical challenges, recent SOTA works (MonoVLN [ICCV 25], Dynam3d [NeurIPS 25]) adopt similar settings. In our 100m² environment, these instructions already involves cross-room navigation with turns and occlusions.The reported 85% vs. 70% performance is averaged over 500 physical runs (100 trials × 5 repetitions). We will include performance breakdowns, confidence intervals, and failure cases. While we have thoroughly ablated the individual contributions of module in simulation (Tab.6), our real-world evaluation focuses on validating the end-to-end system. The consistent +15% gain over a standard open-loop baseline (VLN-3DFF) supports the effectiveness of the closed-loop correction mechanism in practice. In our system (robot + Wi-Fi + remote RTX 4090), inference takes \~0.08s per step (baseline \~0.06s), while real-world step latency is dominated by sensing and motion (\~7.4s vs. \~7.2s). Adaptation runs asynchronously in the background (\~0.8s per update) and does not affect the control loop.
> 5. **Qualitative Analysis.** Thanks for valuable suggestion. We will include failure cases and analysis in the appendix. Failures mainly occur under degraded/ambiguous visual conditions, such as texture-less regions (e.g., blank walls) or highly symmetrical environments (e.g., long corridors). Here, instantaneous observations lack discriminative information, making latent alignment unreliable and feedback signal  $Δ_t$ less informative. This reveals a limitation: the feedback mechanism implicitly assumes sufficiently informative observations at each step. When this assumption is violated, the correction signal may be less effective despite the presence of historical context. A potential direction is to incorporate uncertainty-aware modulation of  $Δ_t$, allowing the model to rely more on prediction when observations are ambiguous.

---

> > ### Author Rebuttal · Reviewer_wcqu · 2026-04-05
> >
> > Thank the authors for their detailed response. My question has been solved.

---

> > > ### Author Response · Authors · 2026-04-07
> > >
> > > We sincerely thank the reviewer for the positive feedback and for acknowledging that the concern has been fully resolved.
> > >
> > > We are glad that our response was helpful in clarifying the questions. We will incorporate the relevant clarifications into the final version to further improve the clarity of the paper.
> > >
> > > We appreciate the reviewer’s time and support.

---

### Decision · Program_Chairs · 2026-04-30

**Decision:**

Accept (regular)

**Comment:**

Generally, reviewers found this paper reasonably technically sound, reasonably well-written, and reasonably novel. It may be useful to at least some fraction of the ICML community.